# CD20/MS4A1 is a mammalian olfactory receptor expressed in a subset of olfactory sensory neurons that mediates innate avoidance of predators

Hao-Ching Jiang[1,2,3,8], Sung Jin Park [1,8], I-Hao Wang [1,2,4], Daniel M. Bear[5,6], Alexandra Nowlan [5,7] & Paul L. Greer [1] ✉

The mammalian olfactory system detects and discriminates between millions of odorants to elicit appropriate behavioral responses. While much has been learned about how olfactory sensory neurons detect odorants and signal their presence, how specific innate, unlearned behaviors are initiated in response to ethologically relevant odors remains poorly understood. Here, we show that the 4-transmembrane protein CD20, also known as MS4A1, is expressed in a previously uncharacterized subpopulation of olfactory sensory neurons in the main olfactory epithelium of the murine nasal cavity and functions as a mammalian olfactory receptor that recognizes compounds produced by mouse predators. While wildtype mice avoid these predator odorants, mice genetically deleted of CD20 do not appropriately respond. Together, this work reveals a CD20-mediated odor-sensing mechanism in the mammalian olfactory system that triggers innate behaviors critical for organismal survival.

To survive, animals must accurately detect, correctly interpret, and appropriately respond to sensory stimuli in their environment. For most non-primate mammals, the richest source of this information is the immense variety of small molecules present in their external surroundings, which may signify the presence of predators, food, or mates[1,2]. These chemicals are primarily detected by olfactory receptors (ORs) expressed at the sensory endings of peripheral olfactory sensory neurons (OSNs), which are coupled to the higher brain circuits tasked with mediating odor perception and initiating olfactory-driven behavior[3–6]. However, how mammals detect and process different classes of olfactory stimuli to initiate distinct behaviors is still not well understood (*i.e.*, how does a mouse know to avoid a cat but to actively seek out a piece of cheese?).

One emerging hypothesis is that distinct subpopulations of OSNs might be responsible for different behaviors. The olfactory system can be subdivided into multiple anatomically and molecularly distinct subpopulations of OSNs. In the mouse, there are at least nine distinct olfactory subsystems, each of which is made up of unique and non-overlapping collections of OSNs[7–15]. A handful of these olfactory subsystems have been extensively studied, which has led to significant insight into their role in olfactory perception and odor-driven behaviors. Particularly important for elucidating the role of these olfactory subsystems has been the identification of the ORs that they express. Each subsystem expresses different types of ORs, which enable them to detect subsets of chemical space and mediate specific behaviors. For instance, the largest subdivision in the mouse, the main olfactory

[1]Program in Molecular Medicine, University of Massachusetts Chan Medical School, Worcester, MA, USA. [2]Morningside Graduate School of Biomedical Sciences, University of Massachusetts Chan Medical School, Worcester, MA, USA. [3]Program in Neuroscience, University of Massachusetts Chan Medical School, Worcester, MA, USA. [4]Interdisciplinary Graduate Program, University of Massachusetts Chan Medical School, Worcester, MA, USA. [5]Department of Neurobiology, Harvard Medical School, Boston, MA, USA. [6]Present address: Wu Tsai Neurosciences Institute, Stanford University, Palo Alto, CA, USA. [7]Present address: Bowles Center for Alcohol Studies, University of North Carolina School of Medicine, Chapel Hill, NC, USA. [8]These authors contributed equally: Hao-Ching Jiang, Sung Jin Park. ✉e-mail: Paul.Greer@umassmed.edu

system, owing to its immense receptor repertoire of ~1000 distinct odorant receptors[3], is able to detect essentially all volatile odorants and therefore plays a key role in odor discrimination and odorant-dependent learning[16–18]. Smaller subsystems, such as the vomeronasal subsystem and the trace amine-associated receptor (TAAR) subsystem, express much smaller receptor repertoires that are more narrowly tuned to recognize specific classes of behaviorally relevant odorants[13,19] and may therefore have more specialized roles in identifying odors of innate significance and initiating specific patterns of unlearned behaviors critical for survival[13,20,21].

Nonetheless, despite progress in elucidating the function of a few of these olfactory subsystems, the specific roles of others remain poorly understood. One of the least understood is the olfactory necklace subsystem, which seems to mediate seemingly opposing behaviors for both feeding and innate avoidance of noxious stimuli[10,22]. Perhaps the biggest hurdle toward understanding the role of the necklace system in odor-driven behavior is that until recently, it was unclear how it detects odorants. We identified the membrane-spanning 4A (MS4A) family of proteins as a novel set of ORs in mammalian necklace OSNs[23]. Heterologous expression of individual MS4A proteins in HEK293 cells conferred the ability to respond to specific chemical compounds. Moreover, the in vitro odor-receptive fields of MS4A proteins matched those of the necklace OSNs in which MS4A proteins were expressed[23]. Nonetheless, an absence of mouse lines in which *Ms4a* gene expression was genetically manipulated meant that the role of MS4A proteins in necklace olfactory function was only examined in in vitro and ex vivo experiments, therefore preventing a rigorous assessment of whether MS4A proteins participate in odor detection in vivo. Indeed, because MS4A proteins do not resemble any previously described olfactory receptors—they are four-transmembrane spanning proteins rather than seven-transmembrane GPCRs, there remains some skepticism about whether MS4A proteins function as ORs in vivo[24]. Here, we use *Ms4a* knockout mice to show that MS4A proteins function as bona fide ORs in vivo. Moreover, we show that the MS4A family member MS4A1 (better known as CD20, a protein previously identified as a co-receptor for the B cell receptor in lymphocytes) is not expressed in the necklace but is instead expressed in a previously undescribed subset of OSNs outside the necklace. Within this subpopulation of OSNs, MS4A1 senses predator odorants leading to innate, unlearned avoidance behaviors.

## Results

### MS4A proteins function as chemoreceptors in necklace OSNs in vivo and mediate specific odor-driven avoidance behaviors

Our previous work suggested that *Ms4a* genes encode a new family of non-GPCR mammalian ORs[23]. However, a lack of genetically modified mice in which *Ms4a* expression could be manipulated precluded a definitive determination of whether *Ms4a* genes do, in fact, encode bona fide ORs that function in vivo. Addressing this issue is particularly important given the unusual structure and expression pattern of MS4A proteins in the mammalian olfactory system[23]. To circumvent potential issues of redundancy between *Ms4a* family members, we took advantage of a mouse in which CRISPR/Cas9 technology was deployed to delete all 17 murine *Ms4a* genes (hereafter referred to as *Ms4a* cluster knockout mice) (Figs. 1A, S1A, and S1B). *Ms4a* cluster knockout mice are viable, fertile, produced at Mendelian frequency, and are overtly indistinguishable from their wildtype littermates, enabling us to assess olfactory performance in these *Ms4a*-deficient animals (Fig. S1C).

To begin to test the role of *Ms4a* genes in olfactory function, we initially exposed freely behaving *Ms4a*-deficient mice or their wildtype littermates to 2,5-dimethylpyrazine (2,5-DMP), oleic acid (OA), or alpha-linolenic acid (ALA), previously described in vitro ligands of MS4A6C, MS4A6D, and MS4A4B, respectively, and measured activation of necklace OSNs, which express MS4A proteins[23], by detecting S6 phosphorylation (pS6), a well-established marker of OSN activation[25].

Deletion of *Ms4a* genes eliminated necklace OSN pS6 responses to each of the MS4A-triggering compounds (Figs. 1B, C, and S1D). By contrast, *Ms4a* cluster knockout mice necklace cells responded without impairment to carbon disulfide (CS$_2$), which is detected by necklace cells through the actions of the receptor guanylate cyclase, GC-D, in an MS4A-independent manner (Fig. 1B and C). Thus, *Ms4a* deletion did not disrupt the health of necklace cells or their capacity to respond to odors in general but instead specifically prevented their detection of MS4A ligands suggesting that MS4A proteins function as ORs in necklace OSNs in vivo.

Next, we wanted to examine what role, if any, MS4A olfactory receptors play in odor-driven behaviors. The necklace olfactory system in which *Ms4a* genes are expressed has been implicated in two distinct MS4A-independent olfactory behaviors—the innate avoidance of CO$_2$ at concentrations above those naturally found within the atmosphere[10] and the social transmission of food avoidance triggered by CS$_2$[22] or the urinary peptides guanylin and uroguanylin[26]. Each of these behaviors is thought to be mediated through the receptor guanylate cyclase, GC-D[22,26,27]. As MS4A receptors are also expressed in necklace OSNs[23], we sought to determine whether MS4A receptors contribute to similar types of innate odor-driven behaviors.

We initially focused our efforts on determining whether MS4A receptors mediate innate, odor-driven avoidance responses as these behaviors are robust, reproducible, do not require prior training, and are easily quantifiable. To begin to determine whether MS4As mediate innate avoidance behaviors, we first tested whether any previously identified MS4A ligands induce avoidance in wild-type mice in an unlearned manner. 2,3-dimethylpyrazine (2,3-DMP), 2,5-DMP, OA, linolenic acid (LA), ALA, and arachidonic acid (AA) all activate MS4A-expressing cells in vitro and necklace OSNs in vivo[23]. 2,3-DMP and 2,5-DMP are found in the urine of wolves and ferrets, natural mouse predators, respectively, and prior work has suggested that these compounds are ethologically relevant odors for mice[28–31]. By contrast, OA, LA, ALA, and AA are long-chain fatty acids found in natural food sources of mice[32,33]. To determine whether any of these MS4A ligands trigger innate avoidance responses, we compared the aversive behavior of *Ms4a* cluster knockout and wildtype mice in response to these compounds and to 2,3,5-trimethyl-3-thiazoline (TMT), a component of fox feces[34] that is not an MS4A ligand. The MS4A ligands 2,3-DMP and 2,5-DMP, as well as the previously described aversive odorant, TMT, induced innate, unlearned avoidance responses in wildtype mice (Figs. 1D, E, and S1E). Although wildtype mice robustly avoided DMP, *Ms4a* cluster knockout mice acted indifferently to the presence of DMP and behaved as though no odor was present (Fig. 1D, E, and S1F). This effect of *Ms4a* deletion on mouse avoidance behavior was specific to DMP; *Ms4a* cluster knockouts exhibited similar aversive behaviors as their wildtype littermates in response to other ethologically relevant aversive odors that are not MS4A ligands, such as TMT (Fig. 1D and E). In addition, the deletion of *Ms4as* did not affect other non-odor-mediated avoidance behaviors, such as the amount of time spent in open arms in an elevated plus maze assay (Fig. S1G). Taken together, these results indicate that *Ms4a* genes encode olfactory receptors that mediate specific odor-driven avoidance responses in mammals.

### Ms4a6c detects DMP in necklace OSNs in vivo but does not fully mediate avoidance behaviors to DMP

We next sought to determine which of the 17 *Ms4a* family members were responsible for mediating the unlearned avoidance mice exhibit in response to DMP. Because DMP is sensed by MS4A6C in vitro[23], we initially focused on this MS4A family member and utilized a mouse line in which the *Ms4a6c* gene was specifically deleted (Fig. S2A and B). Like *Ms4a* cluster knockout mice, *Ms4a6c* knockout mice were viable, fertile, and overtly indistinguishable from their wildtype littermates

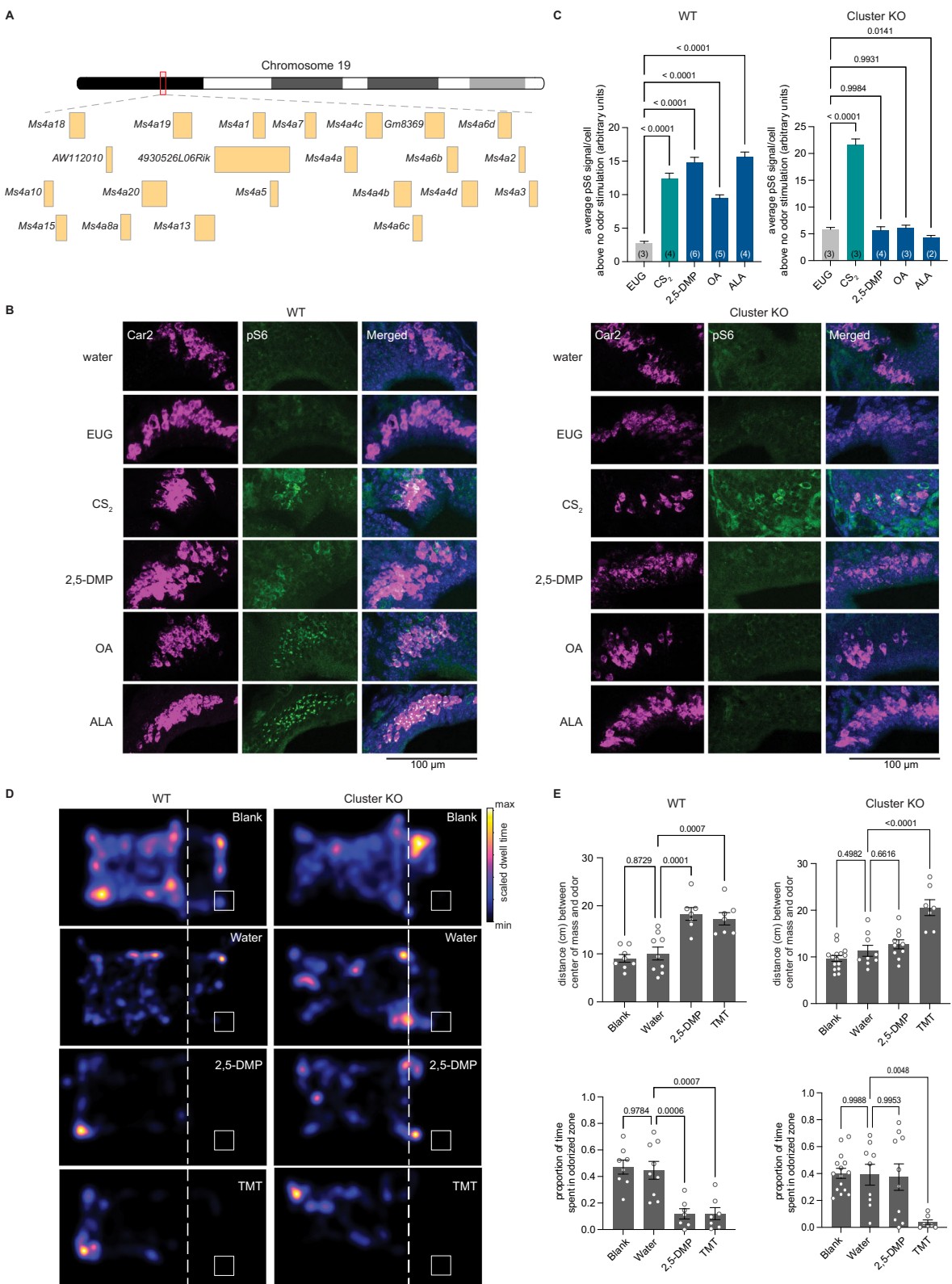

(Fig. S2C and D). Consistent with previous work demonstrating that MS4A6C detects DMP in vitro, *Ms4a6c* knockout necklace neurons did not respond to 2,3-DMP or 2,5-DMP as assessed by pS6 staining (Figs. 2A, B and S2E). By contrast, necklace neurons of *Ms4a6c* knockout mice still robustly responded to OA, the in vitro ligand of the closely related MS4A family member, MS4A6D[23] and to CS₂, a GC-D ligand[22], indicating that *Ms4a6c* deletion specifically impairs the ability

of necklace cells to detect 2,3-DMP and 2,5-DMP and does not generally disrupt their ability to sense non-MS4A6C odorants (Fig. 2A and B).

To determine whether the failure of necklace cells to detect MS4A6C ligands altered the avoidance of these odorants, we assessed innate avoidance responses to 2,5-DMP by *Ms4a6c*-deficient mice. Surprisingly, although *Ms4a6c* knockout mice avoided 2,5-DMP somewhat less than wildtype mice, they avoided 2,5-DMP significantly

**Fig. 1 | Deletion of all members of the MS4A family prevents in vivo detection of MS4A ligands by necklace OSNs and the innate avoidance behaviors triggered by these ligands. A** Schematic representation of the genomic organization of the entire Ms4a family of genes in a tandem array on chromosome 19. **B** Example images of the cul-de-sac regions (where necklace cells reside) of the main olfactory epithelia of mice exposed to the indicated odorant, co-labeled for the necklace cell marker Car2 (magenta) and the neuronal activity marker phospho-S6 (pSerine240/244) (green) from wildtype (left panels) or *Ms4a* cluster knockout animals (right panels). **C** Quantification of the average pS6 signal/necklace cell above the average pS6 signal/necklace cell when no odor is introduced (water) in wildtype mice (left panel) or *Ms4a* cluster knockout mice (right panel) when exposed to the indicated odors. Data are presented as mean ± SEM. For wildtype mice (left panel), $n = 222$ cells over three independent experiments (EUG), $n = 305$ cells over four independent experiments (CS$_2$), $n = 521$ cells over six independent experiments (2,5-DMP), $n = 609$ cells over five independent experiments (OA), $n = 476$ cells over four independent experiments (ALA); for *Ms4a* cluster knockout mice (right panel), $n = 382$ cells over three independent experiments (EUG), $n = 482$ cells over three independent experiments (CS$_2$), $n = 316$ cells over four independent experiments (2,5-DMP), $n = 362$ cells over three independent experiments (OA), $n = 325$ cells over two independent experiments (ALA). The number of independent

experiments (mice) is indicated in brackets, ****$p < 0.0001$, Dunnett's test following one-way ANOVA compared to eugenol (EUG) exposure. **D** Heat maps of the occupancy of wildtype (left panels) or cluster knockout mice (right panels) in the odor avoidance chamber in response to the indicated odorants. Small square represents the location of the odorant, and the dashed line demarcates the odor avoidance zone from the rest of the chamber. Scale bar, 5 cm. **E** Quantification of odor avoidance behavior. The distance between the average center of mass of the mouse and the location of the odorant (top panels) and the proportion of time spent in the odorized zone (bottom panels) were determined for wildtype mice (left) and *Ms4a* cluster knockout mice (right). Each circle represents an individual mouse. Data are presented as mean ± SEM. For wildtype mice (left panels), $n = 8$ mice examined over five independent experiments (Blank), $n = 9$ mice over five independent experiments (Water), $n = 7$ mice over four independent experiments (2,5-DMP), $n = 7$ mice over four independent experiments (TMT); for cluster knockout mice (right panels), $n = 15$ mice over 11 independent experiments (Blank), n= nine mice over seven independent experiments (Water), $n = 10$ mice over six independent experiments (2,5-DMP), $n = 7$ mice over four independent experiments (TMT). **$p < 0.01$, ***$p < 0.001$, Dunnett's test following one-way ANOVA compared to water exposure. Source data are provided as a Source Data file.

---

more than *Ms4a* cluster knockout mice (Figs. 2C, D, S2F, 3E, and S3C). This result suggests that at least one additional *Ms4a* family member may mediate innate avoidance of DMP.

### CD20 responds to DMP and mediates DMP-driven innate avoidance behaviors

To identify additional MS4A receptor(s) that sense DMP, we assessed the ability of all 17 murine MS4A family members to detect 2,5-DMP by detecting DMP-induced calcium responses to odorants in HEK293 cells co-expressing individual *Ms4a* genes with the genetically encoded, fluorescent calcium indicator, GCaMP6s. HEK293 cells do not express endogenous MS4A proteins[23] but exogenously expressed MS4A proteins are efficiently trafficked to the plasma membrane within these cells (Fig. S3A). HEK293 cells expressing either MS4A6C or MS4A1, but none of the other MS4A family proteins, responded to 2,5-DMP to generate a transient calcium signal (Fig. 3A and B), suggesting that MS4A1 is the other MS4A family member mediating the mouse's innate avoidance response to 2,5-DMP. To test this hypothesis, we assessed the ability of *Ms4a1* knockout mice to avoid 2,5-DMP. *Ms4a1* knockout mice acted like *Ms4a* cluster knockout mice—exhibiting no avoidance responses to this predator-derived compound (Figs. 3C–E, S3B and S3C). The failure of *Ms4a1*-deficient mice to respond to 2,5-DMP was specific to this odor since *Ms4a1* knockout mice avoided other aversive odorants, such as TMT, to the same extent as wildtype mice (Fig. 3C and D). Moreover, *Ms4a1* knockout mice were overtly indistinguishable from wildtype mice in other ways—they exhibited similar locomotive behaviors and behaved similarly to wildtype mice in assays of anxiety (such as the elevated plus maze) (Fig. S3D and E), strongly suggesting that the failure to respond to 2,5-DMP was a specific defect in this particular odor-driven behavior and not a sign of more general nervous system dysfunction.

The observation that MS4A1 is required for a mouse to avoid the predator-derived compound 2,5-DMP was surprising since the only previously ascribed function of MS4A1 is as a co-receptor for the B-cell receptor in circulating mature lymphocytes, where it is known as CD20[35,36]. Although it seemed unlikely that lymphocytes would play a critical role in mediating this olfactory-driven behavior, we assessed the ability of *Rag-1*-deficient mice, which lack all mature lymphocytes[37], to avoid 2,5-DMP. *Rag-1* knockout mice avoided DMP to a similar extent as wildtype mice, indicating that mature lymphocyte function was not required for avoidance of 2,5-DMP and further suggesting that CD20 might act in cells outside of the immune system to mediate avoidance of this odor (Fig. S3F and G).

### CD20 is expressed in a previously unidentified subpopulation of OSNs

To identify cells in the olfactory system in which *Ms4a1* might be expressed, we stained coronal sections of the mouse olfactory epithelium with an antibody specific to MS4A1. A relatively sparse population of MS4A1-expressing cells was found, whose cell bodies reside within the epithelial layer of the main olfactory epithelium (MOE) (Fig. 4A). To verify this unexpected observation, we stained coronal sections of the mouse olfactory epithelium with two additional anti-MS4A1 antibodies (raised in different species and recognizing different MS4A1 epitopes). These three anti-MS4A1 antibodies all co-labeled the same cells in the MOE (Fig. 4B). These antibodies did not stain any cells in olfactory epithelial sections obtained from *Ms4a1* knockout mice, confirming their specificity (Fig. 4C). Moreover, combined fluorescent in situ hybridization and immunohistochemistry experiments detected *Ms4a1* mRNA and MS4A1 protein in the same cells indicating that *Ms4a1* is expressed in non-lymphoid cells of the mouse olfactory system (Fig. 4D).

The cell bodies of MS4A1-expressing cells resided in the same anatomic location as OSN cell bodies and extended what appeared to be sensory dendrites to the lumen of the MOE and axonal-like structures toward the olfactory bulb, suggesting that MS4A1-expressing cells might be OSNs. To confirm that MS4A1-expressing cells are neurons, we co-stained for MS4A1 and the neuronal marker NeuN and found that all MS4A1-expressing cells in the olfactory epithelium also expressed NeuN (Fig. 4E). Consistent with this observation, MS4A1 cells did not stain for KI18, a marker of glial support cells[38], KI17, a marker of horizontal basal cells[38], or NeuroD1, a marker of globose basal cells[39] (Fig. 4E). MS4A1-expressing cells expressed CNGA2, a cyclic nucleotide-gated olfactory channel found in mature OSNs[40,41], but not OMP, a marker of conventional GPCR OR-expressing OSNs nor other MS4A family members[42,43] (Figs. 4F and S4A). Together, these results suggest that MS4A1 is expressed in an unconventional neuronal cell type in the olfactory epithelium.

In mammals, a number of distinct subpopulations of olfactory sensory neurons have been previously identified, which are characterized by their unique anatomic and/or molecular features[7–15]. To determine to which, if any, of these olfactory subdivisions, the MS4A1-expressing cells we identified might belong, we performed immunohistochemical and fluorescent in situ hybridization analyses. MS4A1 was not expressed in Gucy1b2-expressing OSNs, TrpC2- or TrpM5-positive OSNs, Pde2a-expressing necklace OSNs (Fig. 4G and H), or OSNs of the vomeronasal organ or Grueneberg Ganglion (Figs. 4H and S4B). Nonetheless, using a combination of iDISCO tissue

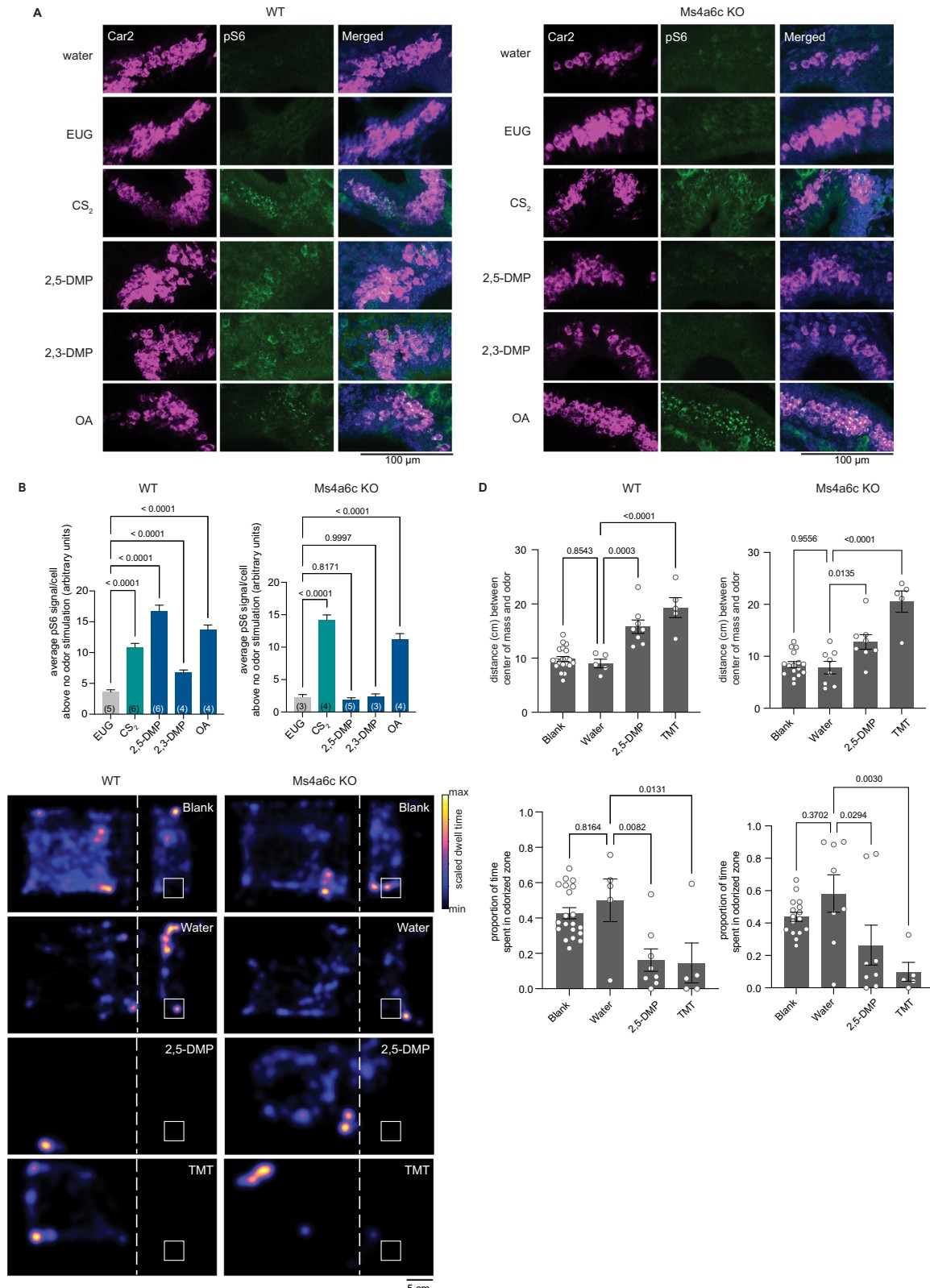

clearing and light sheet microscopy, we found that like other OSN populations, MS4A1-expressing neurons also extended their axons into glomeruli within the mouse olfactory bulb suggesting that MS4A1 is expressed in a previously uncharacterized population of olfactory sensory neurons in the olfactory epithelium and that like other members of the MS4A family, MS4A1 might also function as an olfactory chemoreceptor (Figs. 4I, S4C, and S4D).

## MS4A1 detects nitrogenous heterocyclic compounds in vitro

To begin to test this hypothesis and to explore what types of odors MS4A1 might detect, we examined whether heterologously expressed MS4A1 might respond to additional extracellular chemicals to mediate a calcium influx in HEK293 cells co-expressing GCaMP6s (Fig. 5A). Expression of MS4A1 did not increase the baseline rate of calcium transients in HEK293 cells (Fig. S5A) but increased intracellular calcium

**Fig. 2 | The knockout of *Ms4a6c* impairs the ability of necklace OSNs to detect DMP, a predator-derived compound, but does not significantly affect DMP-mediated innate avoidance behavior. A** Example images of the cul-de-sac regions of the main olfactory epithelia of mice exposed to the indicated odorant, co-labeled for the necklace cell marker, Car2 (magenta), and the neuronal activity marker, phospho-S6 (pSerine240/244) (green), from wildtype (left panels) or *Ms4a6c* knockout animals (right panels). **B** Quantification of the average pS6 signal/necklace cell above the average pS6 signal/necklace cell when no odor is introduced (water) in wildtype mice (left panel) or *Ms4a6c* knockout mice (right panel) when exposed to the indicated odors. Data are presented as mean ± SEM. For wildtype mice (left panel), $n = 360$ cells over five independent experiments (EUG), $n = 404$ cells over six independent experiments (CS$_2$), $n = 459$ cells over six independent experiments (2,5-DMP), $n = 611$ cells over four independent experiments (2,3-DMP), $n = 467$ cells over four independent experiments (OA); for *Ms4a6c* knockout mice (right panel), 269 cells over three independent experiments (EUG), $n = 351$ cells over four independent experiments (CS$_2$), $n = 193$ cells over five independent experiments (2,5-DMP), $n = 281$ cells over three independent experiments (2,3-DMP), $n = 501$ cells over four independent experiments (OA). The number of independent experiments (mice) is indicated in brackets, $**p < 0.01$, $****p < 0.0001$,

Dunnett's test following one-way ANOVA compared to eugenol exposure. **C** Heat maps of the occupancy of wildtype (left panels) or *Ms4a6c* knockout mice (right panels) in the odor avoidance chamber in response to the indicated odorants. Small square represents the location of the odorant, and the dashed line demarcates the odor avoidance zone from the rest of the chamber. Scale bar, 5 cm. **D** Quantification of odor avoidance behavior. The distance between the average center of mass of the mouse and the location of the odorant (top panels) and the proportion of time spent in the odorized zone (bottom panels) were determined for wildtype mice (left) and *Ms4a6c* knockout mice (right). Each circle represents an individual mouse. Data are presented as mean ± SEM. For wildtype mice (left panels), $n = 20$ mice over 13 independent experiments (Blank), $n = 5$ mice over five independent experiments (Water), $n = 8$ over four independent experiments (2,5-DMP), $n = 5$ mice over four independent mice (TMT); for *Ms4a6c* knockout mice (right panels), $n = 16$ mice over nine independent experiments (Blank), $n = 8$ mice over four independent experiments (Water), $n = 10$ mice over five independent experiments (2,5-DMP), $n = 5$ mice over four independent experiments (TMT). $*p < 0.05$, $**p < 0.01$, $***p < 0.001$, $****p < 0.0001$, Dunnett's test following one-way ANOVA compared to water exposure. Source data are provided as a Source Data file.

spikes upon presentation of specific chemicals (Fig. 5A and B). This was true for both human and mouse MS4A1 proteins (Figs. 5A and S5B). MS4A1 responses were tuned to nitrogenous heterocyclic compounds, including 2,3-DMP, 2,5-DMP, and 2,6-DMP, and to a lesser extent, indole and quinoline (Figs. 5B and S5C). However, not all nitrogenous heterocyclic compounds induced calcium transients in MS4A1-expressing cells, nor did non-nitrogenous compounds like isoamyl acetate (IAA) and vanillin, indicating some ligand specificity (Fig. 5B). Dose–response curves revealed nanomolar and low micromolar EC50s for two specific MS4A1-ligand pairs, which is well within the range of what has been observed for other mammalian odorant receptor/ligand relationships (Fig. 5C). Moreover, depletion of extracellular calcium completely eliminated all calcium transients observed in response to ligand presentation (Figs. 5D and S5D). Together, these observations suggest that MS4A1 is a chemoreceptor that detects nitrogenous heterocyclic compounds.

### Nitrogenous heterocyclic compounds activate MS4a1-expressing OSNs in vivo

These experiments were all carried out in vitro, and it remains unclear whether MS4A1 functions as an olfactory receptor in an intact mouse. To test this, freely behaving mice were exposed to the in vitro identified ligands for MS4A1 in the gas phase, and the activation of MS4A1-expressing cells was then assessed. 2,3-DMP and 2,5-DMP both activated MS4A1-expressing OSNs in vivo (Fig. 6A and B). By contrast, ligands for other non-MS4A olfactory receptors, such as eugenol and CS$_2$, did not activate MS4A1-expressing cells above the background (Fig. 6A and B). These experiments reveal that during conditions of active exploration, MS4A1-expressing cells respond to the chemicals we identified as MS4A1 ligands, indicating that MS4A1 functions as an olfactory receptor in vivo.

## Discussion

Here, we took advantage of *Ms4a* knockout mice and used a combination of behavioral experiments and neuronal activation assays to show that MS4A proteins function as ORs within necklace subsystem OSNs in vivo. We also identified an OSN subsystem that expresses MS4A1/CD20, but not other GPCR or MS4A ORs that were probed. MS4A1-expressing OSNs were sparse and located in the MOE and were dispersed rather than geographically localized. We showed that MS4A1 recognizes nitrogenous heterocyclic compounds found in the urine of mouse predators, and their sensing triggers innate, unlearned avoidance behavior.

These experiments convincingly demonstrate the existence of a non-GPCR family of mammalian ORs. All previously identified

mammalian OR families were exclusively seven transmembrane-spanning GPCRs[3]. The discovery that *Ms4a* genes encode a polymorphic set of non-GPCR ORs raises questions about why this family of ORs evolved and what advantages it provides to mammalian olfaction. MS4A chemoreceptors respond to fairly non-descript chemical classes, including nitrogenous heterocyclic compounds, long-chain fatty acids, and steroids that can also be sensed by conventional GPCR ORs[44] (HCJ, SJP, and PLG unpublished observation). This finding suggests that MS4A proteins probably did not evolve to detect chemical compounds that the rest of the olfactory system does not recognize but, rather, more likely evolved to mediate specific types of odor-driven behaviors. This is in line with the observation that in contrast to the "one-receptor, one-neuron" pattern of expression displayed by all other studied mammalian ORs, whereby each OSN expresses one, and only one of the approximately 1200 OR genes encoded by the murine genome[3,45], many different MS4A proteins are co-expressed within the same necklace sensory neuron[23]. This unorthodox pattern of expression suggests that MS4As may be important for mediating specific patterns of behavior rather than for the exquisite discriminatory capacity that the rest of the olfactory system possesses. Interestingly, our experiments suggest that necklace-expressed MS4A proteins, unlike MS4A1, which is expressed outside the necklace, do not play a major role in innate avoidance responses to their ligands. They likely are important in initiating other types of odor-driven behavior that remain to be defined. Perhaps the most likely behaviors induced by the necklace system may be social behaviors since MS4A ligands are enriched for semiochemicals and pheromones[23]. Moreover, necklace neurons have been implicated in the social transmission of food preference[22], a behavior whereby a mouse conveys its prior food experience to a conspecific animal. Thus, it is intriguing to speculate that MS4A proteins may participate in these or related behaviors.

While questions remain about the role of necklace-expressed MS4A members in odor-driven behaviors, this work identifies a function of the non-necklace cell-expressed MS4A family member, MS4A1. Here, we report that *Ms4a1* encodes an olfactory chemoreceptor that is expressed in a previously uncharacterized population of OSNs within the MOE. MS4A1 detects nitrogenous heterocyclic compounds that are found in high abundance in the urine of natural predators of the mouse, such as the wolf and the ferret, and we find that MS4A1 is required for the innate avoidance responses that mice exhibit in response to these compounds. Intriguingly, MS4A1 is expressed in a relatively small population of OSNs in the MOE, and prior work from a number of other laboratories has revealed that other discrete populations of sensory neurons, including

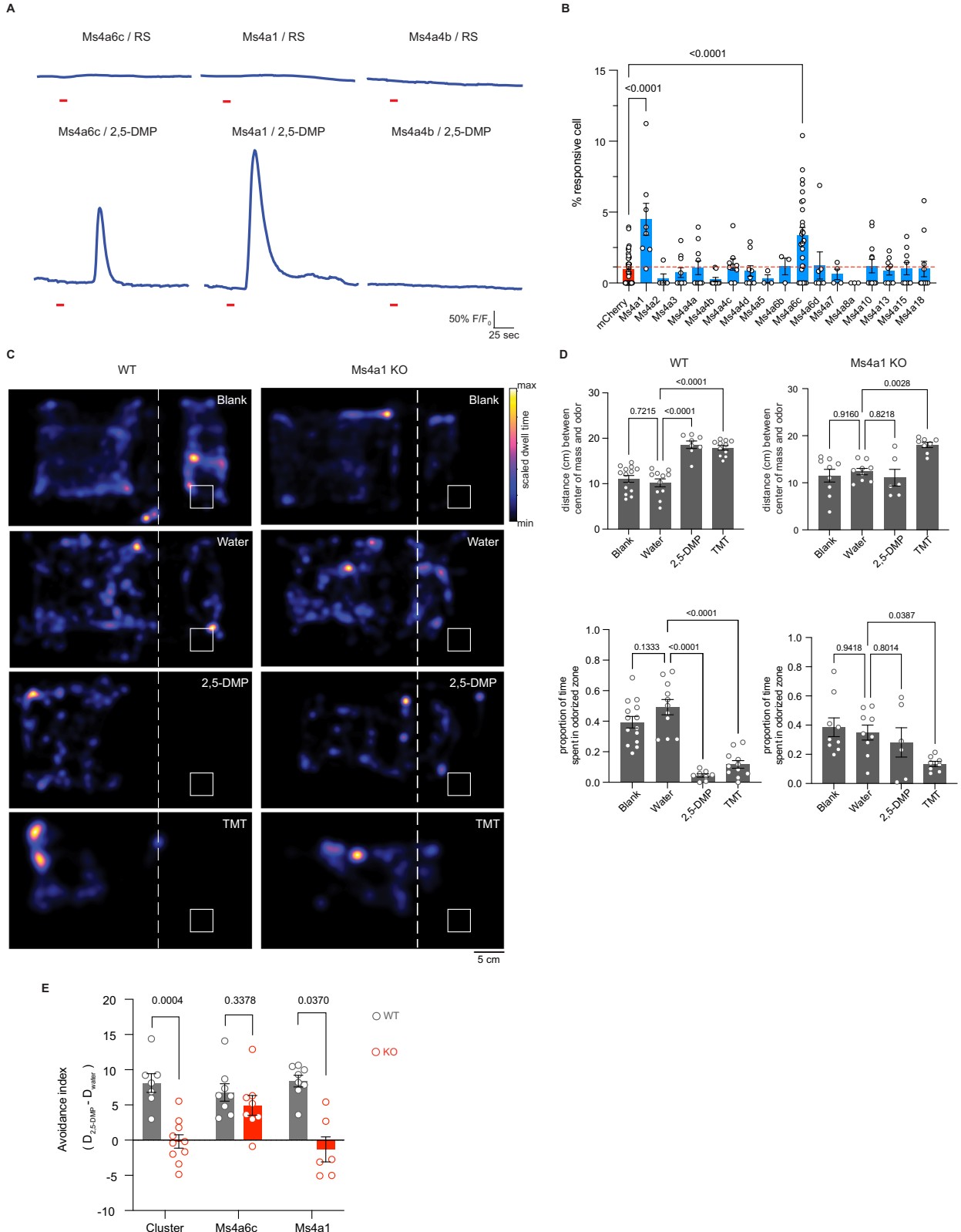

TAAR-expressing cells, necklace cells, and Gruenberg ganglion neurons also mediate innate avoidance responses[10,21,22,46]. Why multiple systems have evolved independently to mediate the innate avoidance of predators remains unknown. Perhaps, the avoidance of predators is such an essential skill to enable the animal to survive that it has evolved a number of redundant systems to execute these tasks. Alternatively, it may be that avoidance behaviors are only triggered when multiple of these systems are activated simultaneously, thus preventing the animal from erroneously running away from odorants it shouldn't. Most of the predator-derived odorants that have been identified to date can also be found in non-predator sources, and a coincident detection requirement might prevent the animal from spending an inordinate amount of time fleeing non-existent threats. Differentiating between these possibilities will likely be facilitated by

**Fig. 3 | MS4A1/CD20 facilitates the detection of DMP, and the deletion of *Ms4a1* eliminates innate avoidance of DMP. A** GCaMP6s fluorescence in response to the indicated chemicals in HEK293 cells expressing the indicated MS4A protein (odor delivery indicated by red bars). RS: 1X Ringer's solution. **B** Quantification of responses of expressed MS4A protein to 2,5-DMP as in (**A**). $n = 39$ wells of cells over 14 independent experiments (mCherry), $n = 8$ wells of cells over three independent experiments (Ms4a1), $n = 5$ wells of cells over three independent experiments (Ms4a2), $n = 10$ wells of cells over three independent experiments (Ms4a3), $n = 10$ wells of cells over three independent experiments (Ms4a4a), $n = 9$ wells of cells over three independent experiments (Ms4a4b), $n = 9$ wells of cells over three independent experiments (Ms4a4c), $n = 9$ wells of cells over three independent experiments (Ms4a4d), $n = 3$ wells of cells over two independent experiments (Ms4a5), $n = 3$ wells of cells over two independent experiments (Ms4a6b), $n = 27$ wells of cells over 13 independent experiments (Ms4a6c), $n = 7$ wells of cells over two independent experiments (Ms4a6d), $n = 4$ wells of cells over two independent experiments (Ms4a7), $n = 3$ wells of cells over two independent experiments (Ms4a8a), $n = 12$ wells of cells over four independent experiments (Ms4a10), $n = 9$ wells of cells over three independent experiments (Ms4a13), $n = 9$ wells of cells over three independent experiments (Ms4a15), $n = 12$ wells of cells over four independent experiments (Ms4a18). Dashed red line indicates the mean plus one standard deviation above the responses of HEK293 cells expressing mCherry alone in response to 2,5-DMP. ****$p < 0.0001$, Dunnett's test following one-way ANOVA compared to mCherry alone. **C** Heat maps of the occupancy of wildtype (left panels) or *Ms4a1* knockout mice (right panels) in the odor avoidance chamber in response to the indicated odorants. Small square represents the location of the odorant, and the dashed line demarcates the odor avoidance zone from the rest of the chamber. Scale bar, 5 cm.

**D** Quantification of odor avoidance behavior. The distance between the average center of mass of the mouse and the location of odorant (top panels) and the proportion of time spent in the odorized zone (bottom panels) were determined for wildtype mice (left) and *Ms4a1* knockout mice (right). Each circle represents an individual mouse. Data are presented as mean ± SEM. For wildtype mice (left panels), $n = 14$ mice over six independent experiments (Blank), $n = 11$ mice over six independent experiments (Water), $n = 8$ mice over four independent experiments (2,5-DMP), $n = 11$ mice over four independent experiments (TMT); for *Ms4a1* knockout mice (right panels), $n = 9$ mice over three independent experiments (Blank), $n = 9$ mice over six independent experiments (Water), $n = 6$ mice over three independent experiments (2,5-DMP), $n = 8$ mice over three independent experiments (TMT). *$p < 0.05$, **$p < 0.01$, ****$p < 0.0001$, Dunnett's test following one-way ANOVA compared to water exposure. **E** An avoidance index was calculated for cluster knockout mice, *Ms4a6c* knockout mice, *Ms4a1* knockout mice, and their wildtype littermate controls by subtracting the average distance in cm between the average position of a mouse from water from the average position of the mouse and 2,5-DMP. A more positive value represents a larger avoidance of DMP. For wildtype mice (gray circles), the avoidance index was calculated for $n = 7$ mice over four independent experiments (Cluster), $n = 8$ mice over four independent experiments (Ms4a6c), $n = 8$ mice over four independent experiments (Ms4a1); for knockout mice (red circles), $n = 10$ mice over six independent experiments (Cluster), $n = 8$ mice over three independent experiments (Ms4a6c), $n = 6$ mice over three independent experiments (Ms4a1). The data are presented as mean ± SEM. Samples from each group were subjected to bootstrapping. A two-tailed $t$-test was subsequently performed on these data to compare wildtype and knockout mice. Source data are provided as a Source Data file.

characterizing how sensory information received by these olfactory subsystems is processed within the brain. Little is currently known about the neural circuitry downstream of these specialized OSN subpopulations that trigger innate avoidance responses, and in the future, elucidating how information flows from MS4A1-expressing neurons (and other olfactory subsystems that trigger innate avoidance) is likely to reveal how odor-driven innate avoidance behaviors are generated. It is intriguing to speculate that perhaps all olfactory circuits that mediate innate avoidance responses converge on a set of neurons that is responsible for driving fear responses (e.g., specific neurons located within the medial amygdala or ventral medial hypothalamus[2,47]), but future experiments will be required to characterize the neural basis by which these sensory cues are translated into behavioral actions.

To fully address these questions, it will be important to characterize all of the different subpopulations of sensory neurons within the mammalian olfactory system. Our identification of a previously undescribed population of OSNs, with a corresponding recently characterized OR, suggests that there are likely still additional populations of OSNs (and ORs) to be found. The fact that MS4A1 falls outside of the canonical GPCR rubric suggests that the traditional means of identifying additional ORs by relying on homology to known ORs may be insufficient. RNA sequencing and spatial transcriptomics will facilitate the identification of additional olfactory subsystems.

This work may also have implications for understanding immune function. The only previously ascribed function for MS4A1 is as a co-receptor for the B-cell receptor in mature lymphocytes[35,36]. The discovery that MS4A1 possesses chemoreceptive properties within the olfactory system suggests that MS4A1 may also function as a chemoreceptor in immune cells. Consistent with this idea, we find that B lymphocyte signaling is also activated by the MS4A1 ligand, 2,5-DMP (Fig. 6C and D). Future work to identify what ligands MS4A1 senses in B lymphocytes and what effects their sensing has on B cell function are likely to be revealing. Other MS4A family members, besides MS4A1, are also found in other cell types and tissues throughout the body, including peripheral immune cells, microglia, reproductive cells, and lung cells[48–51]. Polymorphisms in *Ms4a* genes have been strongly and

reproducibly linked to a number of human diseases not obviously linked to olfaction, including Alzheimer's disease and asthma[52–55]. Therefore, characterizing the role of this family of chemoreceptors in non-olfactory contexts is likely to provide insight into organismal function in both healthy and disease states.

## Methods

### Mice

This study used several mouse lines (mus musculus) following federal guidelines (12 h light/12 h dark cycle, 20–23 °C, 30–70% humidity) and was given food and water ad libitum. *Ms4a* cluster knockout mice were generated in the Datta Lab (Harvard Medical School) by standard approaches using CRISPR/Cas9 technology and will be described in detail elsewhere. *Ms4a6c* knockout mice were generated by KOMP using homologous recombination. These *Ms4a* knockout mice were kindly provided by the Datta Laboratory. *Ms4a1* knockout mice (C57/BL6) were obtained from the Tedder Lab, Duke University[56]. All behavioral and immunostaining experiments with knockout mice were performed with littermate wild-type control mice. The following primer sequences were used for genotyping: (1) *Ms4a* cluster knockout, common primer (5'-GACAAATGAACTAACCTTGCTTGG-3'), wild type-specific primer (5'-TCCAGTGGAAGTGGTTTTTGC-3'), and deletion specific primer (5'-GCCTTGGCTAGGCTACAACC) were used to amplify a fragment of 412 bp from the wildtype allele and 259 bp from the deleted allele. (2) *Ms4a6c* knockout, a 204 bp fragment from the wild type allele was amplified with one primer set (5'-GGACAGAAAC GCCTAAAGGT-3' and 5'-AGAGAAGGGAGATGGTGACTACTA-3'), and a second set of primers (5'-CTAAACTCAAGAGGTCATTGAAG-3' and 5'-GCAGCGCATCGCCTTCTATC-3') amplified a 280 bp fragment from the targeted allele. (3) *Ms4a1* knockout, a 487 bp fragment from the wild type allele was amplified with one primer set (5'-GATATCTAC GACTGTGAACC-3' and 5'-GGCATGTGCCCAGTAAGCC-3'), and a second set of primers (5'-TTGGGGGCTGTCCAAATCATG-3' and 5'-CATCGCC-GACAGAATGCCC-3') amplified a 445 bp fragment from the targeted allele. All mouse husbandry and experiments were performed following institutional and federal guidelines and approved by the University of Massachusetts Chan Medical School's Institutional Animal Care and Use Committee.

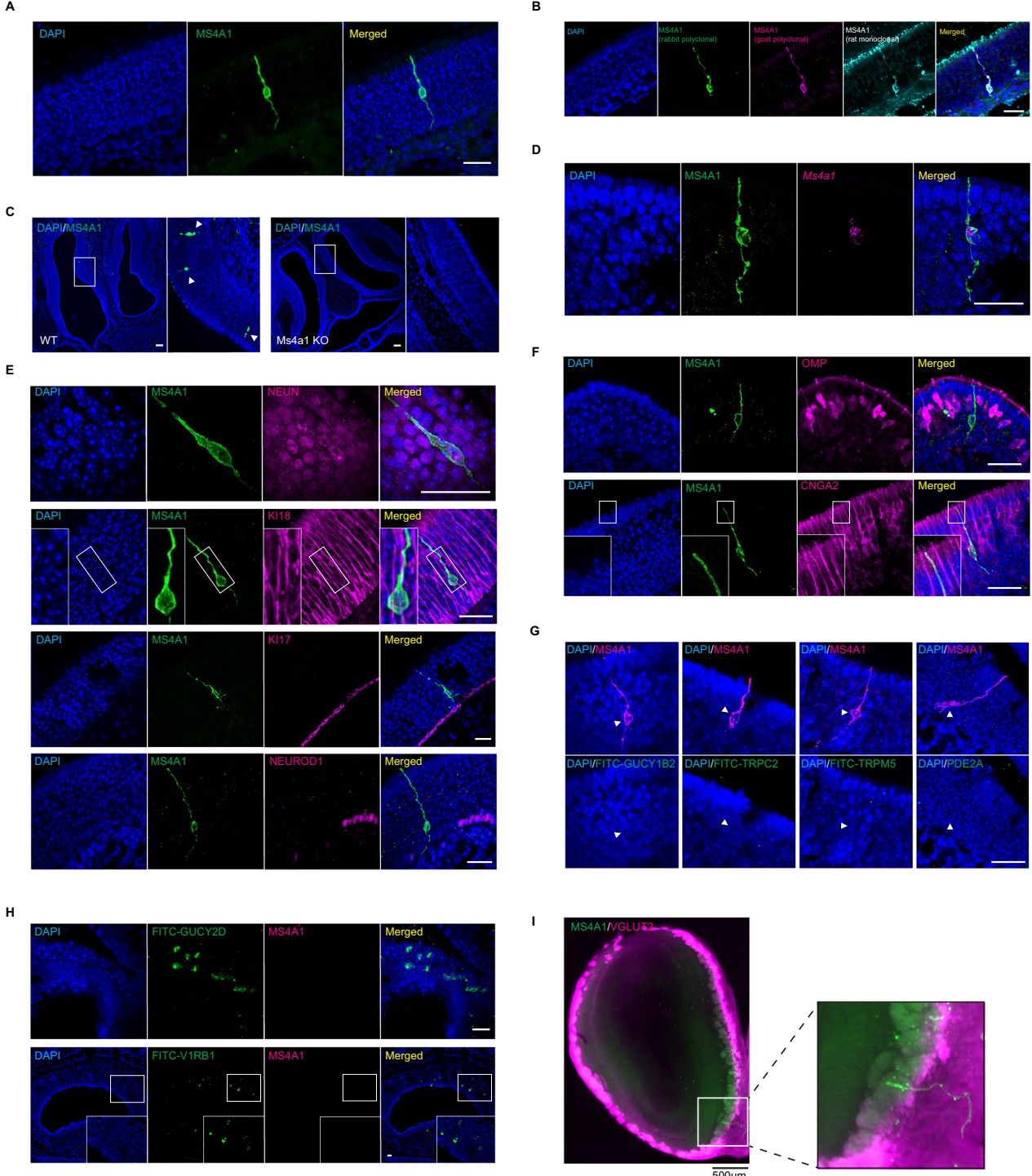

## Plasmids

mCherry was cloned into the pcDNA3.1(+) backbone (mCherry-pcDNA3.1). The complete coding sequences of mouse Ms4a2, Ms4a3, Ms4a4a, Ms4a4b, Ms4a4c, Ms4a4d, Ms4a5, Ms4a6b, Ms4a6c, Ms4a7, Ms4a8a, Ms4a10, Ms4a13, Ms4a15, Ms4a18 were cloned into mCherry-pcDNA3.1. Mouse or human Ms4a1 DNA coding sequences were cloned into the tetracycline-inducible mammalian expression plasmid pcDNA5-FRT-TO. pGP-CMV-GCaMP6s was a gift from Douglas Kim (Addgene, #40753). pISH-Gucy1b2-probe1 (Addgene, #105454), pISH-Gucy1b2-probe2 (Addgene, #105455), pISH-Trpc2-probe1 (Addgene, #105473), pISH-Trpc2-probe2 (Addgene, #105474), pISH-Trpm5

(Addgene, #105993), pISH-Gucy2d-1 (Addgene, #105459), pISH-V1rb1 (Addgene, #16010) were gifts from Peter Mombaerts.

## Antibodies

Primary antibodies/concentrations used were as follows: rabbit anti-phospho-S6 ribosomal protein (Serine240/244) (1:100, Cell Signaling Technologies, #2215), rabbit anti-phospho-S6 ribosomal protein (Serine244/247) (1:150, Invitrogen, #44-923G), rabbit anti-MS4A1/CD20 (1:250 for immunostaining, 1:100 for iDISCO, Cell Signaling Technology, #98708), rabbit anti-MS4A1 (1:200, MyBioSource, #MBS2051903), goat anti-MS4A1 (1:50, Santa Cruz Biotechnology, #sc-7735), rat

**Fig. 4 | MS4A1 is expressed in a previously unidentified subpopulation of OSNs.**
**A** Expression of MS4A1 protein in solitary cells of the main olfactory epithelium. Scale bar, 20 µm. $n \geq 10$ mice were tested. **B** Immunostaining was performed with rabbit polyclonal (green), goat polyclonal (magenta)), and rat monoclonal (cyan) anti-MS4A1 antibodies recognizing different epitopes of the protein. Scale bar, 20 µm. $n = 3$ mice were tested. **C** Immunostaining of MS4A1 in the main olfactory epithelia of wildtype and *Ms4a1* knockout mice. MS4A1-expressing cells are detected in sections obtained from wildtype (indicated by white arrowheads, left panels) but not *Ms4a1* knockout (right panels) mice. Scale bar, 80 µm. $n = 4$ mice were tested for both wild-type and *Ms4a1* knockout mice. **D** Detection of *Ms4a1* mRNA expression (magenta) in anti-MS4A1 antibody labeled cells (green) using combined single-molecule fluorescent in situ hybridization and immunohistochemistry. Scale bar, 20 µm. $n = 3$ mice were tested. **E** Determination of whether MS4A1-expressing cells co-express NeuN (neuronal marker, top panels), KI18 (sustentacular cell marker, second panels from the top), KI17 (horizontal basal cell marker, third panels from the top), and NeuroD1 (globose basal cell marker, bottom panels). Scale bar, 20 µm. $n = 4–6$ mice were tested for each marker. **F** Expression of CNGA2 (lower panels) but not OMP (upper panels) in MS4A1-expressing cells. Scale bar, 20 µm. $n = 4–6$ mice were tested for each marker. **G** MS4A1-expressing cells do not express the genes *Gucy1b2*, *Trpc2*, *Trpm5*, and *Pde2a*, markers of previously described olfactory subsystems. Scale bar, 20 µm. $n = 4–6$ mice were tested for each marker. **H** MS4A1 is undetected in necklace cells, marked by their expression of *Gucy2d* (upper panels), and vomeronasal olfactory neurons, marked by their expression of *V1rb1* (lower panels). Scale bar, 20 µm. $n = 4–6$ mice were tested for each marker. **I** iDISCO immunostaining using antibodies that recognize MS4A1 and VGLUT2, an olfactory glomerular marker, reveals that MS4A1-expressing cells coalesce their axons in the olfactory bulb (left panel). Zoomed-in image of the glomerulus where axons from MS4A1-expressing cells terminate (right panel). $n = 3$ mice were tested for both wildtype and *Ms4a1* knockout mice.

anti-MS4A1 (1:100, LifeSpan Biosciences, #LS-C107163-100), guinea pig anti-VGLUT2 (1:500, SYSY, #135 404), rabbit anti-NeuN (1:500, Abcam, #ab104225), rabbit anti-KI18 (1:500, Abcam, #ab52948), rabbit anti-KI17 (1:500, Abcam, #ab53707), goat anti-NeuroD1 (1:50, R&D Systems, #AF2746), goat anti-OMP (1:1000, Wako Chemicals, #544-10001-WAKO), rabbit anti-CNGA2 (1:200, Alomone Labs, #APC-045), and rabbit anti-PDE2A (1:500, FabGennix, #PD2A-101AP).

Secondary antibodies/concentrations used were as follows: Alpaca anti-rabbit-Alexa488 (1:333, Jackson Immunoresearch, #611-545-215), alpaca anti-rabbit-rhodamine red X (RRX) (1:333, Jackson Immunoresearch, #611-295-215), goat anti-rabbit-Alexa647 (1:333, Invitrogen, #A-21245), bovine anti-goat Alexa488 (1:333, Jackson Immunoresearch, #805-545-180), bovine anti-goat Alexa647 (1:333, Jackson Immunoresearch, #805-605-180), goat anti-rat Alexa488 (1:333, Invitrogen, #A-11006), donkey anti-rat RRX (1:333, Jackson Immunoresearch, #712-295-153), and goat anti-guinea pig Alexa647 (1:333, Invitrogen, #A21450).

## Odors
Eugenol, CS₂, 2,3-DMP, 2,5-DMP, OA, ALA, indole, quinoline, pyridine, pyrrolidine, vanillin, and IAA were purchased from Sigma-Aldrich. TMT was purchased from BioSRQ. All odor compounds were obtained at the highest purity possible.

## Odor exposure for phospho-S6 immunostaining
8–12-week-old mice, including *Ms4a* cluster knockout, *Ms4a6c* knockout, *Ms4a1* knockout, and littermate wild-type control mice, were individually acclimated to clean plastic cages (Innovive, # M-BTM) for at least 16 h before the start of experiments. Before introducing odors to the mice, the mice were fasted for 2 h. To initiate the experiments, water or odor stimuli (eugenol, CS₂, 2,3-DMP, 2,5-DMP, OA, ALA) were introduced into each cage. The stimuli were applied by placing 150 µL of water or odorant on filter paper (Sigma-Aldrich, #WHA10347509) in 35 mm Petri dishes. After 2 h of exposure to the odor, the mice were euthanized, and nasal epithelial sections were collected.

## Tissue slice preparation
Mice were euthanized according to our IACUC protocol using CO₂ and secondary cervical dislocation, and their noses, including the olfactory epithelia and attached olfactory bulbs, were dissected from the skull. The dissected tissue was fixed overnight in 4% paraformaldehyde (PFA, Electron Microscopy Sciences, #15714) in phosphate-buffered saline (PBS) at 4 °C. After washing three times for 5 min with 1X PBS, noses were decalcified overnight at 4 °C in 0.45 M EDTA in PBS. Subsequently, the noses were sequentially incubated in 10%, 20%, and 30% sucrose in PBS (Sigma-Aldrich, #S0389) overnight at 4 °C. Finally, the tissues were embedded in Tissue Freezing Medium (Tissue-Tek, #4583). Cryosections of 20-micron thickness were cut onto Superfrost Plus glass slides (VWR #48311-703) and stored at −80 °C until staining.

## Combined pS6 immunostaining and RNAscope fluorescent in situ hybridization (FISH)
For *Car2* and *Ms4a1* RNA detection, RNAscope FISH was performed on nasal epithelial sections from 8–12-week-old C57/BL6 wild-type, *Ms4a* cluster knockout, *Ms4a1* knockout, and *Ms4a6c* knockout mice that were exposed to water or odors as described above. The staining protocol followed the guidelines provided in the Advanced Cell Diagnostics RNAscope Multiplex Fluorescent Reagent Kit v2 Assay User Manual (323100-USM) without the target retrieval step, and all required reagents were obtained from RNAscope Multiplex Fluorescent Detection kit v2 (Advanced Cell Diagnostics, #323110).

Frozen sections were thawed at room temperature for 10 min, then were fixed with 4% PFA in PBS for 15 min at 4 °C. The sections were dehydrated with 50%, 70%, and 100% ethanol for 5 min each at room temperature (RT). The sections were treated with hydrogen peroxide for 10 min at RT, then washed with MilliQ water three times. Sections were treated with protease III at 40 °C in a hybridization oven (HybEZ oven, Advanced Cell Diagnostics, #310010) for 30 min, then washed with MilliQ water three times. Subsequently, the sections were hybridized with either the 3-Plex positive control RNA probe, the 3-Plex negative control RNA probe, the Car2 RNA probe, or the Ms4a1 RNA probe in a 1:50 ratio for 2 h at 40 °C in the oven.

To amplify the hybridization signal, the section slides underwent incubation with three different amplifiers: AMP1 for 30 min, AMP2 for 30 min, and AMP3 for 15 min, all at 40 °C in the oven. After the amplification steps, slides were treated with horseradish peroxidase (HRP) for 15 min at 40 °C in the oven. Following this, the sections were incubated with diluted TSA plus Cy-3 (1:750, PerkinElmer, #NEL741001KT) for 30 min at 40 °C in the oven, then the sections were incubated with HRP blocker for 15 min at 40 °C in the oven. Washing was performed with RNAscope wash buffer (2 min twice at RT) between each step following probe hybridization. The mouse target probes used in this study were as follows: Ms4a1-c1 (#318671-C1), Ms4a1-c2 (#318671-C2), Ms4a4b-c2 (#314611-C2), Ms4a6c-O1-c3 (#435341-C3), and Car2-c2, (#313781-C2).

For the pos-RNAscope immunostaining, sections were blocked with blocking buffer (0.1% Triton X-100 (Sigma-Aldrich, #X100) 5% Normal Donkey serum (Jackson ImmunoResearch, #017-000-121), 3% Bovine Serum Albumin (VWR, #97061-416) in PBS) for 30 min at RT. Sections were then incubated with anti-pS6 antibodies (1:100) in a blocking buffer overnight at 4 °C. On the following day, the slides were washed three times with PBS (5 min each at RT) and then incubated with the secondary antibody (1:300) in a blocking solution for 45 min at RT. Afterward, the slides were washed three times with PBS (5 min each at RT) and mounted using Vectashield antifade mounting media with DAPI (Vector Laboratories, #H-1200-10). To secure the coverslips, nail polish was applied, and the slides were imaged using confocal microscopy, following the procedures described below.

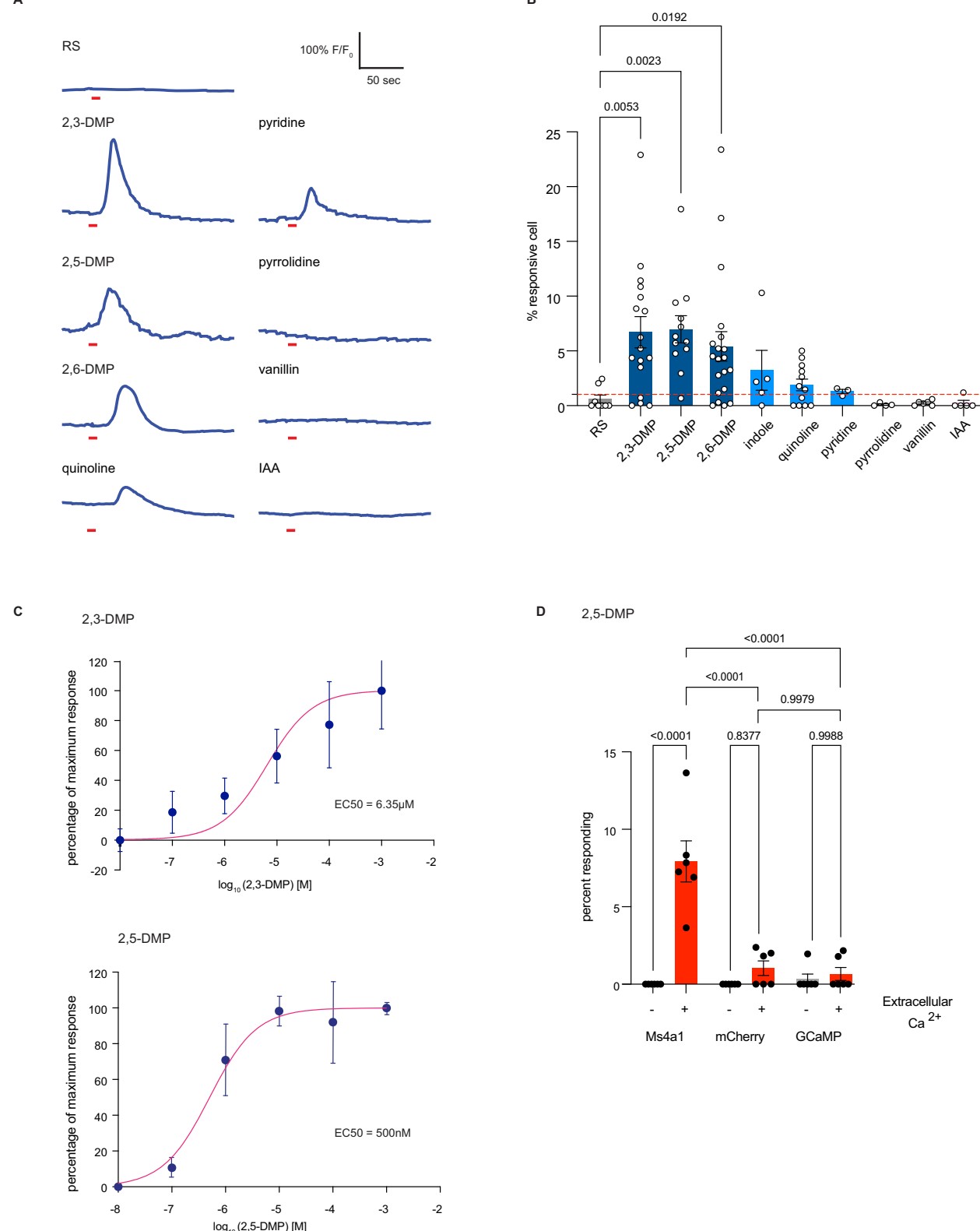

## Conventional FISH

For Gucy1b2, Trpc2, Trpm5, Gucy2d, and V1rb1 RNA detection, conventional FISH was performed on nasal epithelial sections from C57/BL6 wild-type mice, following a modified protocol[57]. The RNA probes for these genes were previously described[7,8,58,59]. Fluorescein isothiocyanate (FITC, Roche, #11685619910)-labeled riboprobes were generated through in vitro transcription from fully linearized and purified template plasmids (described in Plasmids) containing target gene sequences using equilibrated phenol (Sigma-Aldrich, #P9346)−chloroform/isoamyl alcohol (Sigma-Aldrich, #25666). Frozen sections were air-dried, fixed with 4% PFA/1X PBS for 10 min, and then acetylated with a mixture of 0.1 M triethanolamine (Sigma-Aldrich, #90279) and 0.25% acetic anhydride (Sigma-Aldrich, #320102) for 15 min at RT. Pre-hybridization was performed in a prehybridization solution

**Fig. 5 | MS4A1 is a chemoreceptor that detects nitrogenous heterocyclic compounds. A** GCaMP6s fluorescence in response to the indicated chemicals in HEK293 cells expressing MS4A1 protein (odor delivery indicated by red bars). **B** Quantification of responses of cells expressing MS4A1 protein to the indicated chemicals as in (**A**). The data are presented as mean ± SEM. $n = 8$ wells of cells over three independent experiments (RS), $n = 17$ wells of cells over nine independent experiments (2,3-DMP), $n = 12$ wells of cells over 11 independent experiments (2,5-DMP), $n = 20$ wells of cells over ten independent experiments (2,6-DMP), $n = 12$ wells of cells over six independent experiments (quinoline), $n = 5$ wells of cells over four independent experiments (indole), $n = 3$ wells of cells over two independent experiments (pyridine), $n = 4$ wells of cells over three independent experiments (pyrrolidine), $n = 6$ wells of cells over three independent experiments (vanillin), $n = 5$ wells of cells over three independent experiments (IAA). Dashed red line indicates the mean plus one standard deviation of responses of MS4A1-expressing HEK293 in response to RS. *$p < 0.05$, **$p < 0.01$, Dunnett's test following Brown-Forsythe and Welch ANOVA tests compared to control stimulation with Ringer's solution (RS) alone. **C** Dose–response curves reveal low micromolar/high nanomolar EC$_{50}$s for MS4A1/2,3-DMP (top panel) and MS4A1/2,5-DMP (bottom panel). For 2,3-DMP (top panel), each data point represents the mean ± SEM from $n = 8$ wells of cells over five independent experiments ($10^{-8}$ M), $n = 10$ wells of cells over six independent experiments ($10^{-7}$ M), $n = 9$ wells of cells over six independent experiments ($10^{-6}$ M), $n = 12$ wells of cells over six independent experiments ($10^{-5}$ M), $n = 6$ wells of cells over six independent experiments ($10^{-4}$, $10^{-3}$ M); For 2,5-DMP, each data point represents the mean ± SEM from $n = 4$ wells of cells over four independent experiments ($10^{-8}$ M), $n = 3$ wells of cells over three independent experiments ($10^{-7}$ M), $n = 4$ wells of cells from four independent experiments ($10^{-6}$, $10^{-5}$, $10^{-4}$, $10^{-3}$ M). **D** The requirement of extracellular calcium for MS4A1 ligand responses was assessed by stimulating HEK293 cells co-expressing GCaMP6s and either MS4A1 or mCherry or GCaMP6s alone with 2,5-DMP in the presence or absence of extracellular calcium. Data are presented as mean ± SEM from $n = 6$ wells of cells over three independent experiments (for each indicated condition). ****$p < 0.0001$, Tuckey's test following one-way ANOVA compared to no extra-cellular calcium. Source data are provided as a Source Data file.

(10 mM Tris, pH 7.5, 600 mM NaCl, 1 mM EDTA, 0.25% SDS, 1X Denhardt's (Sigma, #D-2532), 50% formamide (Roche, #1814320), 300 μg/ml yeast tRNA (Sigma-Aldrich, #R6750)) for 5 h at 65 °C. Following pre-hybridization, the sections were hybridized overnight at 60 °C with FITC-labeled RNA probes (1 μg/ml) in a hybridization solution (10 mM Tris, pH 7.5, 200 mM NaCl, 5 mM EDTA, 0.25% SDS, 1X Denhardt's, 50% formamide, 300 μg/ml yeast tRNA, 10% dextran sulfate (Bio Basic, #DB0160), 5 mM NaH$_2$PO$_4$, 5 mM Na$_2$HPO$_4$).

On the subsequent day, the slides were sequentially washed with the following buffers: 5X standard saline citrate buffer (Invitrogen, #AM9765) (10 min, 65 °C), 50% formamide/1X SSC (30 min, 65 °C), TNE buffer (10 mM Tris, pH 7.5, 0.5 M NaCl, 1 mM EDTA) (20 min, 37 °C), 2X SSC (20 min, 65 °C), and 0.2X SSC (20 min, 65 °C) twice. Quenching of endogenous peroxidase activity was performed using 1% H$_2$O$_2$ (Sigma-Aldrich, #216763)/1X PBS for 15 min at 4 °C, followed by blocking with a blocking buffer (0.1 M Tris, pH 7.5, 100 mM maleic acid (Sigma-Aldrich, #M0375), 150 mM NaCl, 0.1% Tween-20, 2% blocking reagent (Roche, #11096176001), 10% heat-inactivated sheep serum (Equitech-Bio, #SS32-0100)) for 30 min at RT. The samples were then incubated overnight at 4 °C with anti-fluorescein-POD (1:2000, Roche, #11426346910).

On the third day, the slides were washed three times with PBST (1X PBS/0.1% Tween-20), incubated with diluted TSA plus fluorescein (1:50) for 5–10 min at RT, and washed five times with PBST. Finally, the sections were immunostained with anti-MS4A1 antibodies and imaged using confocal microscopy, following the procedures described below.

### Immunostaining
Sections were incubated in a blocking solution containing 5% normal donkey serum, 0.1% Triton-X100, and 1X Tris-buffered saline (TBS) for 1 h at RT. Subsequently, sections were incubated overnight at 4 °C with primary antibodies diluted in a blocking solution. On the following day, slides were washed three times with TBST (0.1% Triton-X100 in TBS) and then incubated with secondary antibodies in a blocking solution for 1 h at RT. Afterward, the slides were washed three times with TBST, counterstained, and mounted using Vectashield antifade mounting media with DAPI. To secure the coverslips, nail polish was applied, and the slides were imaged using confocal microscopy, following the procedures described below.

### Confocal microscopy
Slides were imaged using an LSM 900 Airyscan2 confocal microscope (Zeiss) equipped with various objective lenses, including ×10/0.45 M27, ×20/0.8 M27, ×40/1.1 water Corr M72, and ×63/1.4 oil DIC. To enhance image quality, acquired digital images were processed by applying a median filter to remove debris that was significantly smaller than the structures being analyzed. Additionally, multi-channel Z-stacks were projected into two dimensions using Zen Blue 3.1 software (Zeiss).

### Quantification of phospho-S6 (pS6) positive cells
Car2 or Ms4a1 positive cells were selected using ImageJ software. pS6 intensity was determined following subtraction of background signal from cells in the olfactory epithelium lacking Car2 or Ms4a1 signal. For necklace cells, 12 sections from the posterior olfactory epithelium were collected, and three Car2-positive regions were randomly selected from each section to perform quantification. For Ms4a1-positive cells, 24 sections equally spaced throughout the anterior to posterior axis of the olfactory epithelium were collected, and all the Ms4a1-positive cells from these sections were analyzed. Analysis was performed blinded to genotype and stimulus to ensure unbiased quantification.

### iDISCO
iDISCO was performed on the olfactory bulbs of 8–12-week-old C57/BL6 wild-type mice following the protocol described by Renier and colleagues[60] (online protocol: http://idisco.info/idisco-protocol). Initially, mice were euthanized according to our IACUC protocol using CO$_2$ and secondary cervical dislocation, and their olfactory bulbs were dissected and fixed for 2 h in 4% PFA in PBS at 4 °C. The samples were then washed three times with PBS for 30 min each at RT and then dehydrated using a series of methanol solutions (20%, 40%, 60%, 80%, 100%, Sigma-Aldrich, #179337-4X4L) for 1 h each at RT. Subsequently, the samples were incubated overnight in a mixture of 66% dichloromethane (Sigma-Aldrich, #270997) and 33% methanol at RT and washed twice with methanol. The samples were then bleached using 5% H$_2$O$_2$/ methanol solution at 4 °C overnight. Following bleaching, the samples were rehydrated using a series of methanol solutions (80%, 60%, 40%, 20%) and PBS for 1 h each at RT.

The samples were incubated in permeabilization solution (0.2% Triton X-100, 0.3 M glycine (Merck, #G5417), 20% DMSO (Corning, #25-950-CQC) in PBS) at 37 °C overnight and then placed in blocking solution (0.2% Triton X-100, 6% donkey serum, 10% DMSO in PBS) at 37 °C overnight. Subsequently, the samples were washed in PTwH buffer (0.2% Tween-20, 10 μg/ml heparin (Sigma-Aldrich, #H3393) in PBS) overnight and incubated with primary antibodies diluted in PTwH buffer supplemented with 5% DMSO and 3% donkey serum at 37 °C for 3 days. The samples were extensively washed in PTwH buffer for 1 day and then incubated with secondary antibodies diluted in PTwH buffer with 3% donkey serum at 37 °C for 3 days. Finally, the samples were washed in PTwH buffer overnight before the clearing and imaging process. For clearing, the bulb samples were dehydrated using a series of methanol solutions (20%, 40%, 60%, 80%, 100%, 100%) for 1 h each

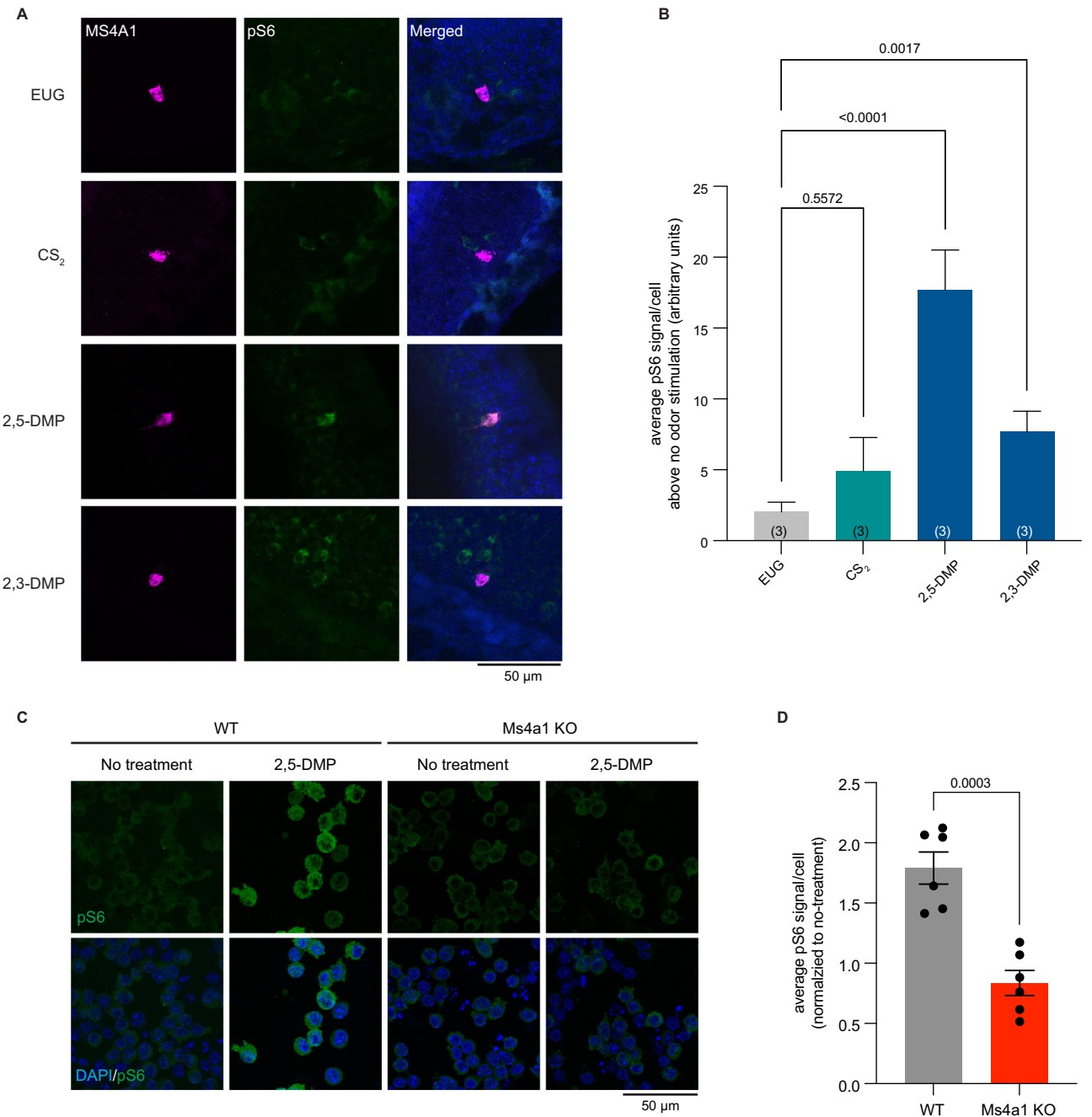

**Fig. 6 | The MS4A1 ligands, 2,3-DMP and 2,5-DMP, activate MS4A1-expressing cells in vivo. A** Example image of the main olfactory epithelia of mice exposed to the indicated odorant, immunostained for the neuronal activity marker, phospho-S6 (pSerine244/247) (green). *Ms4a1* is detected by fluorescent in situ hybridization (magenta), see experimental procedures. **B** Quantification of the average pS6 signal/MS4A1-expressing cell above the average pS6 signal/ MS4A1-expressing when no odor is introduced (water) in wild-type mice when exposed to the indicated odors. Data are presented as mean ± SEM. *n* = 34 cells over three independent experiments (EUG), *n* = 50 cells over three independent experiments (CS$_2$), *n* = 77 cells over three independent experiments (2,5-DMP), *n* = 66 cells over three independent experiments (2,3-DMP). The number of independent experiments (mice) is indicated in brackets, **$p < 0.01$, ****$p < 0.0001$, Dunnett's test following Brown–Forsythe and Welch ANOVA tests compared to eugenol exposure. **C** Example images of 2,5-DMP stimulated wildtype (left panels) or *Ms4a1* knockout (right panels) A20 cells, a BALB/c mouse B cell lymphoma cell line, immunostained for the activity marker phospho-S6 (pSerine240/244) (green). **D** Quantification of the normalized pS6 signal in response to 2,5-DMP in wildtype and *Ms4a1* knockout A20 cells. Data are presented as mean ± SEM from six independent experiments. **$p < 0.01$, two-tailed *t*-test with Welch's correction. Source data are provided as a Source Data file.

at RT. The samples were incubated in 66% dichloromethane/33% methanol at RT for 3 h and then incubated in 100% dichloromethane until they sank to the bottom of the vial. Finally, the samples were incubated in dibenzyl ether (Sigma, #108014) at RT overnight. The cleared samples were directly imaged using a light-sheet microscope (Ultramicroscope II; LaVision BioTec). The images were acquired using InspectorPro software (LaVision BioTec), and three-dimensional

reconstruction and analysis were performed using Imaris ×64 software (v.8.0.1, Bitplane).

**Odor-driven behavior assay**

For behavioral experiments, 8–12-week-old *Ms4a* cluster knockout, *Ms4a6c* knockout, *Ms4a1* knockout, and littermate wild-type control mice were group housed in the behavioral assay room and allowed to

acclimate for at least one day prior to the start of the experiments. Two hours prior to the behavioral assay, mice were individually fasted in their home cage. During the experiment, mice were placed in single-use, disposable cages (Innovive, #M-BTM) with a disposable paper curtain separating the avoidance zone from the odorized zone. Only the odorized zone was enclosed by an acrylic sheet on the top of the cage. A clean filter paper was placed in a 35 mm Petri dish within the odorized zone. Without any odor stimulus, mice were first allowed to freely explore their surroundings for 30 min. Subsequently, the mice were exposed to water (40 μL) and applied to the filter paper in the odorized zone for a duration of 3 min. After water exposure, the same mouse was then exposed to 40 μL of odorant for 3 min. The odorant was delivered onto a fresh filter paper for the experiment. Animal behavior during the entire experiment, including habituation, water exposure, and odor exposure, was recorded with a webcam (Logitech, #LOWCC920S). Those videos were then analyzed with ezTrack[61].

### Elevated plus maze assay
The elevated plus maze (EPM) apparatus consisted of plus-shaped (+) apparatus with two open and two enclosed arms. The closed arms are enclosed by black walls ($30 \times 5 \times 15$ cm), and the open arms are exposed ($30 \times 5 \times 0.25$ cm). The maze was elevated 45 cm above the floor, and a red fluorescent light was positioned 1 m above the maze serving as a light source. The whole assay was performed in a darkroom. 8–12-week-old *Ms4a* cluster knockout, *Ms4a6c* knockout, *Ms4a1* knockout, and littermate wild-type control mice were group housed in the darkroom for 1 h prior to the experiment. Individual mice were placed at the center of the maze, and the mouse was allowed to freely explore the maze for 5 min. The time the mice stayed in the open arm and the closed arm is automatically measured by the system. The maze was cleaned with rubbing alcohol prior to and between experiments. The maze was designed by Andrew Tapper[62].

### RNA-Sequencing
Three 8–12-week-old *Ms4a* cluster knockout, *Ms4a6c* knockout, and littermate wild-type control mice were euthanized according to our IACUC protocol using $CO_2$ and secondary cervical dislocation, and the main olfactory epithelium was dissected and immersed quickly in 0.6 ml of ice-cold Trizol (Invitrogen, #15596018). Olfactory epithelia were ground and homogenized with nuclease-free disposable pestles (Fisher Scientific, #12-141-368) in an Eppendorf tube (Eppendorf, #2231000347). The homogenized sample was incubated for 5 min at room temperature. 0.1 μL of chloroform was added to the sample and mixed by inverting for 20 s. The sample was then incubated for 3 min at RT. The sample was centrifuged at 12,000×*g* for 15 min at 4 °C. The clear aqueous phase was collected into a nuclease-free Eppendorf tube, and an equal volume of 70% ethanol (Sigma-Aldrich, #E7023) was added and mixed. The sample was then loaded into the RNeasy spin column, and RNA was extracted with an RNeasy mini kit (Qiagen, #74104). A NEBNext Ultra II Directional RNA Library Prep Kit for Illumina (NEB, #E7490S, and #E7760S) was used to construct sequencing libraries following the manufacturer's guidelines with one alteration, which was to increase the insert length to approximately 300 bp. Libraries were sequenced using paired-end 150-cycle reads on a Novaseq6000 (Novogene). The sequencing reads were processed using the DolphinNext RNAseq pipeline (https://dolphinnext.umassmed.edu/index.php?np=1&id=732)[63]. The default settings were employed, except that STAR v.2.6.1 and RSEM v.1.3.1 were used for alignment and quantification, respectively[64,65]. Transcriptome build: gencode M25. The count matrix was loaded to R (v.4.0.0 or later), and DEseq2 (v.1.30.1) was used to normalize the matrix and perform differential gene expression analysis[66].

### Cell culture and transfection
Human embryonic kidney 293 (HEK293) cells (ATCC, #CRL-3216) were cultured in complete media, which consisted of DMEM-high glucose (Sigma-Aldrich, #D5671) supplemented with 10% heat-inactivated fetal bovine serum (R&D Systems, #S12450H), penicillin/streptomycin/glutamine (Gibco, #10378016), and maintained in a 5% $CO_2$ humidified tissue culture incubator at 37 °C. For calcium imaging experiments, HEK293 cells were transfected with calcium phosphate. $0.5 \times 10^6$ cells were seeded in each well of a 12-well plate, and 1 μg of GCaMP6s and 1 μg of either mCherry or mCherry-MS4A fusion plasmid DNA were mixed with 250 mM $CaCl_2$. The solutions were resuspended by pipetting four times and combined with 2X HBS (containing 50 mM HEPES, 10 mM KCl, 12 mM D-glucose, 280 mM NaCl, and 1.5 mM $Na_2PO_4$ at pH 7.06). The reaction mixtures were incubated for 5 min at RT and then added dropwise to each well. For the expression of MS4A1 protein, 1 h after transfection, tetracycline (Sigma-Aldrich, #T7660) was added to a final concentration of 1 μg/ml. For surface immunostaining, HEK293 cells were plated on 12 mm round German glass coverslips (Bellco Biotechnology, #1943-10012A) coated with poly-d-lysine hydrobromide (Sigma-Aldrich, #P0296) in a 24-well plate and incubated in complete media. The cells were transfected using the same calcium phosphate method as described above.

### Generation of lentiviral CRISPR/CAS9-mediated Ms4a1 knockout B cell lines and pS6 immunostaining
To produce lentivirus, HEK293T cells were transfected with pLenti-CRISPR v2 plasmids containing guide sequences targeting the control gene (ACTATCATGGCACCCAATTG) and the *Ms4a1* gene (GATGG GTGCGAAGACCCCTG)[67], along with delta-Vpr packaging plasmid and the VSV-G envelope plasmid. X-tremeGENE 9 Transfection reagent (Roche, #XTG9-RO) was used for the transfection. Lentivirus-containing media were collected 24 h after changing to fresh media. The supernatant containing the virus was used without concentration after one cycle of freeze and thaw to eliminate any residual HEK293T cells. The virus was then transduced into A20 cells (ATCC, #TIB-208), a mouse BALB/c B cell lymphoma line[68], with 10 μg/ml polybrene for 1 h using centrifugation at RT. The cells were incubated for 24 h, and the media was replaced with media containing 1 μg/ml puromycin to select for transduced cells. Single-cell clones were selected, expanded, and subjected to Western blot analysis to generate single-cell-derived *Ms4a1* knockout A20 cells. For the pS6 activation experiments, both control and *Ms4a1* knockout cells on coverglass were incubated in serum-free media for 30 min and treated with either vehicle or 2,5-DMP for 30 min in a tissue culture incubator at 37 °C. Subsequently, the cells were fixed with 4% PFA/PBS for 20 min at RT, followed by three washes with PBS. The cells were incubated in a blocking buffer for 30 min at RT and then with anti-pS6 antibodies (1:400) in a blocking buffer overnight at 4 °C. On the following day, the cells were washed three times with PBS and then incubated with a secondary antibody (1:300) in a blocking solution for 45 min at RT. After three washes with PBS, the cells were mounted using Vectashield antifade mounting media with DAPI.

### Calcium imaging
24 h post-transfection, media in the wells were aspirated and washed twice with 1X Ringer's solution supplemented with 1 mM $CaCl_2$ ($Ca^{2+}$-Ringer's solution). The wells were then incubated for 30 min in the cell culture incubator with $Ca^{2+}$-Ringer's solution. Following the incubation, the plate was transferred to a Lionheart LX Automated microscope (BioTek), and calcium imaging was performed using Gen5 software (BioTek). Preliminary images were acquired with brightfield, RFP, and GFP filters at 20X magnification prior to each experiment, focusing on an imaging field containing cell numbers between 50 and 200. Subsequently, using the same field of view and fixed *z*-axis, images were captured for 10 min (1 FPS). During the kinetic image acquisition, either $Ca^{2+}$-Ringer's solution as a negative control or specific odorants (50 μM for 2,3-DMP, 2,5-DMP, 2,6-DMP, indole, quinoline, pyridine, pyrrolidine, vanillin, IAA) solubilized in $Ca^{2+}$-Ringer's

solution were pipetted into the upper edge of each well after 360 s for a duration of 10 s. To determine dose–response curves and calculate the EC50, HEK293 cells co-expressing GCaMP6s and either mCherry or mCherry-MS4A1 were treated with six logarithmic orders of 2,3-DMP or 2,5-DMP (ranging from 10 nM to 1 mM) starting with the lowest concentration. For experiments conducted without extracellular calcium, all solutions were replaced with 1X Ringer's solution supplemented with 1 mM EGTA to chelate calcium. All acquired images were aligned to the first image of each experiment using Gen5 software, and subsequent images were analyzed using Fiji software. It is important to note that only a low percentage of Ms4a-expressing cells respond to each odorant, and we have not yet figured out why this is the case. It does not appear to be dependent on either receptor expression or membrane trafficking and likely reflects some aspect of MS4A receptor signaling that has not yet been elucidated.

### Analysis of calcium imaging data

Mp4 videos were converted into a sequence of PNG images with ffmpeg software, the image sequence was then imported into Fiji. GCaMP6s and mCherry positive cells were selected, and their GCaMP6s intensities were calculated across the whole image sequence, which was subsequently analyzed using a customized R script. Briefly, for each selected cell, the average intensity and standard error of GCaMP6s 30 s prior to ligand presentation were calculated. 2.5-fold of the standard error above mean intensity was then used as a threshold to determine if the cell responded to the odor.

### Statistics and reproducibility

For quantification of pS6, at least three biological repeats were performed for each odorant treatment. All analyses were conducted blinded to genotype and stimulus to ensure unbiased quantification. One-way ANOVA was performed to calculate the statistical difference in mean intensity of pS6 from all groups. A post-hoc Dunnett's test was used to determine if the mean intensity of pS6 from a given odorant treatment was significantly different from the eugenol control group.

For the odor-driven behavior assay, at least five biological repeats were performed for each odorant. The analysis was conducted in an automated manner whenever possible in the absence of human supervision to ensure blinding to genotype and stimulus. For the total distance traveled, an unpaired Welch $t$-test was performed to calculate statistical differences. For the distance between the mouse's center of mass and the odor, a paired $t$-test was performed to calculate statistical differences. For comparing the proportion of time spent in the odorized zone and the distance between the mouse's center of mass and odor (unpaired), a one-way ANOVA was performed to calculate statistical differences between all groups. A post-hoc Dunnett's Test was applied to determine if the values from a given odorant treatment were significantly different from those from the water control group. For the avoidance index, an unpaired Welch $t$-test was used to compare each knockout group and their wild-type littermates. A false-discovery rate was controlled using a two-stage step-up developed by Benjamini, Krieger, and Yekutieli.

For the EPM assay, at least five biological repeats were performed for each genotype. An unpaired Welch $t$-test was used to compare time spent in the open arm between groups.

For calcium imaging, at least nine biological repeats were performed for each odorant. The analysis was conducted blinded to the protein expressed and stimulus to ensure unbiased quantification. For identifying 2,5-DMP responsive MS4A receptors, a one-way ANOVA was performed using a post-hoc Dunnett's test. For screening chemicals that might activate MS4A1-expressing cells, a one-way ANOVA was performed with a post-hoc Dunnett's test to compare to cells exposed to Ringer's solution only. For assessing whether the presence of extracellular calcium affects the response rate of MS4A1-expressing cells, a one-way ANOVA was performed with a post-hoc Tukey's test.

For all the symbols indicating statistical significance in this article: ****$p < 0.0001$; ***$p < 0.001$; **$p < 0.01$; *$p < 0.05$; ns, $p \geq 0.05$.

### Reporting summary

Further information on research design is available in the Nature Portfolio Reporting Summary linked to this article.

## Data availability

All RNA sequencing data described in this manuscript are deposited at GEO accession under accession code GSE240378, which is associated with Figs. S1B and S2B. Source data are provided with this paper.

## Code availability

All the scripts used for this study can be found at: https://github.com/Greerlab/CD20_2023_paper. (https://doi.org/10.5281/zenodo.10822757).

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

## Acknowledgements

We thank Judy Lieberman, Eric Greer, Jin Zhang, Bob Datta, and members of the Greer Lab for their helpful comments on the manuscript. We would like to thank Bob Datta and Thomas Tedder for generously providing mouse lines, Rubing Zhao-Shea for technical assistance with the elevated plus maze experiments, and Namgyu Lee and Leonid Yurkovetskiy for helping to generate *Ms4a1* knockout A20 cell lines. P.L.G. was supported by fellowships from the Smith Family Foundation, the Searle Scholars Program, the Rita Allen Foundation, the Whitehall Foundation, and by grants DP2 OD027719-01 and NIH 5 KL2 TR001455-04 from the National Institutes of Health. H.C.J. and I.H.W. were supported by Mello Fellowships. S.J.P. was supported by the Mogam Science Fellowship. D.M.B. is a Biogen Fellow of the Life Sciences Research Foundation and an Interdisciplinary Scholar of the Wu Tsai Neurosciences Institute at Stanford University.

## Author contributions

H.-C.J., S.J.P. and P.L.G. conceived the project and designed the experiments. H.-C.J. and S.J.P. conducted most of the experiments under the supervision of P.L.G. I.-H.W. performed the RNA sequencing experiments and the iDisco experiments. D.M.B. performed pilot experiments with MS4A1, and A.N. worked with P.L.G. to generate *Ms4a* knockout mice.

## Competing interests

The authors declare no competing interests.
