## [Peer Review File · Nature Communications]

CD20/MS4A1 is a mammalian odorant receptor expressed in a subset of olfactory 2 sensory neurons that mediates innate avoidance of predatorsREVIEWER COMMENTS

Reviewer #1 (Remarks to the Author):

In the submitted manuscript, Jiang et al. describe the role of MS4A olfactory receptors in innate avoidance behaviors in rodents. First, the authors show that several *in vitro* agonists of MS4A receptors activate MS4A-expressing neurons *in vivo*. They further show that such activation is abolished in the MS4A cluster KO. Further, behavioral avoidance of one of these odors, 2,5-DMP, is abolished in the MS4A cluster knockout. The authors then show that responses to this odor (and a related one) in necklace neurons depend on the Ms4a6c receptor specifically. However, behavioral responses to the odor are not fully abolished in Ms4a6c knockouts. The authors then identify Ms4a1 as an additional receptor for 2,5-DMP and related odors, and show that behavioral responses are abolished in the Ms4a1 knockout. Interestingly, the authors then report expression of Ms4a1 in sparse neurons distributed in the main olfactory epithelium, identifying a new subtype of olfactory sensory neuron in this tissue. The authors go on to show that this subtype of OSNs indeed responds to 2,5-DMP and related odors in a Ms4a1-dependent manner *in vivo*.

This is a very interesting paper that answers an important question – are the MS4As *in vivo* olfactory receptors? The experiments are authoritatively designed and the data are quite convincing. I think this will be an important step forward in the field. I only have some relatively minor comments:

1. Do the Ms4a1-expressing neurons in the MOE co-express any other Ms4a's? While I do not consider this experiment essential for publication, it would be interesting to know whether the multi-receptor expression pattern found for MS4As holds in this new cell type.
2. I think the Discussion might be enhanced by a mention of potential downstream circuits that could receive inputs from the Ms4a1-expressing neurons.
3. In Fig. 5D, there is a very mild odor response in a few mCherry-only neurons. I'm guessing this is just spurious, but since it's an important control, it may be worth recording a few more for this odor and the other odors (to ensure that the HEK readout is Ms4a1-specific).
4. Fig. 6D is missing statistical analysis.
5. For Fig. 1C, I had to go fishing in the Methods to find out about the data are quantified. I'd suggest a more descriptive y-axis label and a bit more detail in the figure legend on how images are processed for quantification. In addition, it'd be nice to have a zoomed-in panel of a few pS6-expressing neurons in the supplement, as the images in the main figure are fairly zoomed out, making it hard to assess single-cell pS6 immunostaining.
6. Fig. S1B and related panels are missing a legend for the color scale bar.

Reviewer #2 (Remarks to the Author):

The mammalian olfactory system includes several subsystems, and has the ability to detect a variety of odors and trigger distinct innate behaviors. Among the olfactory receptor families in mammals, the MS4A family is unique. Unlike the typical G-protein coupled olfactory receptors, MS4A receptors are not GPCRs and primarily found in the necklace olfactory subsystem. The senior author had previously unveiled these fascinating chemoreceptors.

Building on that foundational research, this manuscript delves into determining the *in vivo* function of MS4As using CRISPR-mediated cluster and single gene knockout mice. The authors highlighted the critical role of an MS4A member, MS4A1, in innate avoidance behavior against 2,5-dimethylpyrazine. Surprisingly, MS4A1 is an exception in its family; it is not expressed in necklace olfactory neurons, but rather in a previously unknown subpopulation of OSNs in the MOE.

In summary, this research marks a significant advance in the field of olfaction. The authors identified a

unique non-canonical olfactory receptor vital for triggering innate avoidance behaviors with rigorous technical approach to address core research questions.

There are some minor points that the authors need to address.

1. The title that appears to claim a totally new odorant receptor does not seem to be justified. While the authors use "CD20" in the title, they use "MS4A1" in the main body .
2. The bar charts representing pS6+ quantification (Fig. 1C, 2B, and 6B) should also include data for water, to reflect the information in the images showing cells labeled with Car2 and phosphoS6 (Fig. 1B, 2A, and 6A).
3. In Fig. 1E, 2D, 3D, and S3E, while the use of the distance (cm) between the center of mass and the odor is convenient, it does not seem to be an appropriate method as a part of non-odorized zone is closer to a part of odorized zone.
4. In Fig. 3B, individual data points constituting each bar graph should be added.
5. Fig 4I is unclear. The authors should show where within the olfactory bulb the MS4A1-expressing cells terminate their axons.
6. In Fig. 5C, large error bars are concerning. The author could mitigate this by showing 95%.
7. 2,5-DMP also activates Grueneberg Ganglion. Does Grueneberg Ganglion express MS4A1 or other MS4A members?

Reviewer #3 (Remarks to the Author):

Jiang et al. builds on this lab's earlier 2016 discovery that the MS4A protein family is implicated in olfactory function in rodents. This study demonstrates that MS4A proteins function in specialized olfactory receptor neurons, such as the necklace ORNs, as odorant receptors *in vivo*, responding to ethologically relevant chemicals including predator and food odors. Genetic loss-of-function of all MS4A family members results in loss of avoidance behavior to 2,5-DMP, an odorant present in carnivore predator urine. Interestingly, this study discovered that one family member, MS4A1/CD20, previously characterized as a BCR co-receptor, functions as an odorant receptor in specialized ORNs in the main olfactory epithelium. MS4A1 appears to respond broadly to nitrogenous heterocyclic molecules, including DMPs, resembling other MS4A members (e.g. MS4a6c). Genetic loss of MS4A1 reduces behavioral avoidance to 2,5-DMP.

The technical quality of the data is strong; for instance, the specificity of reagents such as OR-specific antibodies were validated against genetic null mutants. The reproducibility and rigor of the data is overall very high. In addition, the results presented in this manuscript are substantive and significant and provide important and definitive genetic evidence that MS4A proteins, which are not GPCRs, function *in vivo* as bona fide odorant receptors in the mouse. The study also makes several intriguing observations that set the stage for future discovery of the role of this specialized OR family in the rodent olfactory system, including the identity of MS4A1 as a marker for a putative new class of specialized ORNs linked to innate avoidance.

We have a few questions for the authors:

1. In all three of the MS4A mutant lines, distributions of the proportion of time spent in the 2,5-DMP odorized zone appear like they may be bimodal (Figure 1E, 2D, 3D-E). Does this distribution reflect individual variability/intrinsic spatial biases in animals' trajectories in the absence of stimuli? A figure

showing comparisons between water and DMP within each animal could be interesting to see (as Fig S1 D/3E). Given that the underlying population may not be normally distributed, bootstrap/resampling statistics (ala Efron) may provide a better analysis than Welch's or paired t-test.

2. We are curious if complete loss of MS4A genes (the cluster KO) would lead to a loss of response to natural predator odor, which is a mixture of many monomolecular compounds. Have the authors tested how MS4A cluster knockouts behave towards predator urine?

3. Could the authors comment on why response rates are relatively low (<5%) for MS4A1 and MS4A6C transfected cells? Does this correlate with expression levels or cell-surface trafficking efficiency?

4. Regarding characterization of MS4A1 expression, could the author provide information regarding the location of the MS4A1 glomerulus in the OB? (lines 282-88, Figure 4I). Without spatial references, it's hard to orient. Do MS4A1 axons converge onto a single glomerulus or several glomeruli? Zone I or II?

5. We would be curious to read more about the authors' ideas on why many specialized subsystems (VNO, TAARs, MS4A, etc) have evolved for mediating innate behaviors like predator avoidance. Do the authors expect distinct behaviors will segregate into different systems? Are they redundant? Is there evidence that the rodent olfactory system has evolved multiple receptors more often for ethologically important odors to confer sensory and behavioral robustness/degeneracy?

We would like to thank all three reviewers for their thoughtful comments on our manuscript. We are all too aware of the many responsibilities that they have, and we are grateful to them for taking the time to provide us feedback on our study. We are pleased that all three reviewers viewed our study so positively, and we have attempted to address all of the concerns that they raised in our revised manuscript. In doing so, we feel like we have significantly strengthened our manuscript. Please find a point-by-point response to each of their comments embedded below:

Reviewer #1 (Remarks to the Author):

In the submitted manuscript, Jiang et al. describe the role of MS4A olfactory receptors in innate avoidance behaviors in rodents. First, the authors show that several *in vitro* agonists of MS4A receptors activate MS4A-expressing neurons *in vivo*. They further show that such activation is abolished in the MS4A cluster KO. Further, behavioral avoidance of one of these odors, 2,5-DMP, is abolished in the MS4A cluster knockout. The authors then show that responses to this odor (and a related one) in necklace neurons depend on the Ms4a6c receptor specifically. However, behavioral responses to the odor are not fully abolished in Ms4a6c knockouts. The authors then identify Ms4a1 as an additional receptor for 2,5-DMP and related odors, and show that behavioral responses are abolished in the Ms4a1 knockout. Interestingly, the authors then report expression of Ms4a1 in sparse neurons distributed in the main olfactory epithelium, identifying a new subtype of olfactory sensory neuron in this tissue. The authors go on to show that this subtype of OSNs indeed responds to 2,5-DMP and related odors in a Ms4a1-dependent manner *in vivo*.

This is a very interesting paper that answers an important question – are the MS4As *in vivo* olfactory receptors? The experiments are authoritatively designed and the data are quite convincing. I think this will be an important step forward in the field. I only have some relatively minor comments:

1. Do the Ms4a1-expressing neurons in the MOE co-express any other Ms4a's? While I do not consider this experiment essential for publication, it would be interesting to know whether the multi-receptor expression pattern found for MS4As holds in this new cell type.

We have performed *in situ* hybridization experiments to address this question. These experiments have failed to reveal the expression of other Ms4a genes within Ms4a1-expressing cells. These results are now included as Figure S4A in the revised manuscript. This observation suggests that in contrast to other Ms4a-expressing cells, these do not have the same multi-receptor expression pattern.

2. I think the Discussion might be enhanced by a mention of potential downstream circuits that could receive inputs from the Ms4a1-expressing neurons.

We appreciate the suggestion and have now added this to the discussion. This can be found beginning at line 406 of the revised manuscript.

3. In Fig. 5D, there is a very mild odor response in a few mCherry-only neurons. I'm guessing this is just spurious, but since it's an important control, it may be worth recording a few more for this odor and the other odors (to ensure that the HEK readout is Ms4a1-specific).

In response to this suggestion, we have performed additional experiments in response to both 2,5-DMP and 2,3-DMP. These experiments are now presented as Figures 5D and S5D.

4. Fig. 6D is missing statistical analysis.

This has been rectified in the revised manuscript. We thank the reviewer for bringing this to our attention.

5. For Fig. 1C, I had to go fishing in the Methods to find out about the data are quantified. I'd suggest a more descriptive y-axis label and a bit more detail in the figure legend on how images are processed for quantification. In addition, it'd be nice to have a zoomed-in panel of a few pS6-expressing neurons in the supplement, as the images in the main figure are fairly zoomed out, making it hard to assess single-cell pS6 immunostaining.

We have altered the Y-axis labels in Figures 1C, 2B, and 6B, as suggested as well as including more detail in the figure legends about how the images were processed. We have also included a zoomed in panel of representative pS6 staining as Figures S1D and S2E in the revised manuscript.

6. Fig. S1B and related panels are missing a legend for the color scale bar.

We have added legends for the color scale bars for all such figures. They may be found in Figures S1B and S2B in the revised manuscript.

Reviewer #2 (Remarks to the Author):

The mammalian olfactory system includes several subsystems, and has the ability to detect a variety of odors and trigger distinct innate behaviors. Among the olfactory receptor families in mammals, the MS4A family is unique. Unlike the typical G-protein coupled olfactory receptors, MS4A receptors are not GPCRs and primarily found in the necklace olfactory subsystem. The senior author had previously unveiled these fascinating chemoreceptors.

Building on that foundational research, this manuscript delves into determining the in vivo function of MS4As using CRISPR-mediated cluster and single gene knockout mice. The authors highlighted the critical role of an MS4A member, MS4A1, in innate avoidance behavior against 2,5-dimethylpyrazine. Surprisingly, MS4A1 is an exception in its family; it is not

expressed in necklace olfactory neurons, but rather in a previously unknown subpopulation of OSNs in the MOE.

In summary, this research marks a significant advance in the field of olfaction. The authors identified a unique non-canonical olfactory receptor vital for triggering innate avoidance behaviors with rigorous technical approach to address core research questions.

There are some minor points that the authors need to address.

1. The title that appears to claim a totally new odorant receptor does not seem to be justified. While the authors use "CD20" in the title, they use "MS4A1" in the main body.

We have now changed the title to reflect both CD20 and MS4A1 to make it less confusing. We feel like this will allow people familiar with the role of MS4A proteins in olfaction as well as CD20 in the immune system to both access these results as we feel like they will be of interest to both groups. We thank the reviewer for their suggestion.

2. The bar charts representing pS6+ quantification (Fig. 1C, 2B, and 6B) should also include data for water, to reflect the information in the images showing cells labeled with Car2 and phosphoS6 (Fig. 1B, 2A, and 6A).

We are sorry that this wasn't clear in the original submission. Reviewer #1 had similar difficulty in understanding this figure obviously highlighting that we did a poor job in explaining the experiment in our original submission. We have now altered those figures to make it clearer what the analysis was. We have changed the Y-axis label as well as providing more information in the figure legend. We hope that these changes will satisfy the reviewers' concerns.

3. In Fig. 1E, 2D, 3D, and S3E, while the use of the distance (cm) between the center of mass and the odor is convenient, it does not seem to be an appropriate method as a part of non-odorized zone is closer to a part of odorized zone.

We agree with the reviewer that there are some caveats with using the distance metric for measuring avoidance, particularly given that some parts of the non-odorized zone are closer to the odorized zone than other parts. However, despite this, we think that this metric still provides value and provides a complementary approach to the amount of time spent in the odorized zone. We have been careful to run a number of controls with wildtype mice and known aversive odorants (many of which are found within the manuscript (e.g., Figures 1E, 2D, and 3D) which reveal a robust relationship between distance and aversion. In addition, similar metrics have been published previously (e.g., Perez-Gomez et al., 2015 Current Biology) for assessing odorant avoidance, and in aggregate we therefore feel comfortable including these data and that they provide further support for our claims. However, if the reviewer feels strongly about this, we would definitely consider removing these analyses.

4. In Fig. 3B, individual data points constituting each bar graph should be added.

We have included this in the revised manuscript as suggested.

5. Fig 4I is unclear. The authors should show where within the olfactory bulb the MS4A1-expressing cells terminate their axons.

We have rectified this in Figures 4I and S4C and D in the revised manuscript.

6. In Fig. 5C, large error bars are concerning. The author could mitigate this by showing 95%.

We have performed additional replicates of this experiment to reduce the error bars. These new data are reflected in the revised manuscript in Figure 5C.

7. 2,5-DMP also activates Grueneberg Ganglion. Does Grueneberg Ganglion express MS4A1 or other MS4A members?

We have stained GG for MS4A1 and found that MS4A1 is not expressed in this olfactory organ. This is consistent with our prior paper where we found that the necklace MS4A proteins are not expressed in GG (Greer et al., Cell 2016). These data may now be found as Figure S4B in the revised manuscript.

Reviewer #3 (Remarks to the Author):

Jiang et al. builds on this lab's earlier 2016 discovery that the MS4A protein family is implicated in olfactory function in rodents. This study demonstrates that MS4A proteins function in specialized olfactory receptor neurons, such as the necklace ORNs, as odorant receptors in vivo, responding to ethologically relevant chemicals including predator and food odors. Genetic loss-of-function of all MS4A family members results in loss of avoidance behavior to 2,5-DMP, an odorant present in carnivore predator urine. Interestingly, this study discovered that one family member, MS4A1/CD20, previously characterized as a BCR co-receptor, functions as an odorant receptor in specialized ORNs in the main olfactory epithelium. MS4A1 appears to respond broadly to nitrogenous heterocyclic molecules, including DMPs, resembling other MS4A members (e.g. MS4a6c). Genetic loss of MS4A1 reduces behavioral avoidance to 2,5-DMP.

The technical quality of the data is strong; for instance, the specificity of reagents such as OR-specific antibodies were validated against genetic null mutants. The reproducibility and rigor of the data is overall very high. In addition, the results presented in this manuscript are substantive and significant and provide important and definitive genetic evidence that MS4A proteins, which are not GPCRs, function in vivo as bona fide odorant receptors in the mouse. The study also makes several intriguing observations that set the stage for future discovery of the role of this specialized OR family in the rodent olfactory system, including the identity of

MS4A1 as a marker for a putative new class of specialized ORNs linked to innate avoidance.

We have a few questions for the authors:

1. In all three of the MS4A mutant lines, distributions of the proportion of time spent in the 2,5-DMP odorized zone appear like they may be bimodal (Figure 1E, 2D, 3D-E). Does this distribution reflect individual variability/intrinsic spatial biases in animals' trajectories in the absence of stimuli? A figure showing comparisons between water and DMP within each animal could be interesting to see (as Fig S1 D/3E). Given that the underlying population may not be normally distributed, bootstrap/resampling statistics (ala Efron) may provide a better analysis than Welch's or paired t-test.

The reviewer is astute in pointing this out. To address this concern, we have taken the reviewer's suggestion and analyzed the amount of time each individual mouse spent in the odorized zone when presented with either no odor (blank) or 2,5-DMP. These data reveal that every wildtype mouse spent less time in the odorized zone when 2,5-DMP was present compared to blank. By contrast, for both *Ms4a1* knockout and *Ms4a* cluster knockout mice the difference between control and DMP is random and looks similarly to if you compare the responses to blank and water. For *Ms4a6c* knockout mice, the individual mouse responses to DMP do appear to be somewhat bimodal and probably reflect the partial effect we observe for *Ms4a6c* knockout on DMP avoidance and reflects a partial contribution of this gene to this innate behavior. We have included all of these analyses in the revised manuscript as Figures S1F, S2F, and S3B. Because of these observations, we have subjected the data to a bootstrapping analysis as suggested by the reviewer, and these new statistical analyses are included in the revised manuscript.

2. We are curious if complete loss of MS4A genes (the cluster KO) would lead to a loss of response to natural predator odor, which is a mixture of many monomolecular compounds. Have the authors tested how MS4A cluster knockouts behave towards predator urine?

We agree with the reviewer that this would be of significant interest. During the revision process we have attempted these experiments, but unfortunately, we have failed to observe robust avoidance with purchased wolf urine. There does seem to be a significant difference between the response of wildtype and *Ms4a* cluster knockout mice to wolf urine where cluster knockout mice are more attracted to wolf urine (WP, see embedded figure), but given that we fail to see the expected response to wolf urine we are reticent to include these data as we feel like they are difficult to interpret.

In speaking with our colleagues, it seems that the best source of these urines is from direct collection/collaboration with zoos or similar facilities, and we are in the process of trying to establish these relationships, but we have been unable to perform these experiments within the timeframe of this review process. We agree with the reviewer that these experiments will be of interest in the future.

3. Could the authors comment on why response rates are relatively low (<5%) for MS4A1 and MS4A6C transfected cells? Does this correlate with expression levels or cell-surface trafficking efficiency?

This is another interesting observation by the reviewer. We are not precisely sure why the *in vitro* response rates are so low for MS4A-expressing cells. We have observed similarly low response rates for all MS4A receptors that we have assessed (see Greer et al., Cell 2016). Although the responses are low, they are receptor and ligand-dependent, and they match precisely what we observe *in vivo* (which are absent in *Ms4a* knockout mice), indicating that these responses accurately reflect the chemoreceptive properties of these receptors. We have tried numerous approaches to boost these responses, including making stable cell lines, trying other cell lines, etc.... None of these has yet led to more robust responses and the response rates don't seem to correlate with either expression level or cell-surface trafficking. We think that there is likely some aspect of Ms4a signaling that we do not yet understand, and we are actively working to try and elucidate the mechanisms by which MS4A proteins signal. Unfortunately, to date, we have been unable to crack this problem. We have added a statement about this to the revised manuscript in the methods section as I'm sure other readers will have similar questions.

4. Regarding characterization of MS4A1 expression, could the author provide information regarding the location of the MS4A1 glomerulus in the OB? (lines 282-88, Figure 4I). Without spatial references, it's hard to orient. Do MS4A1 axons converge onto a single glomerulus or several glomeruli? Zone I or II?

We have rectified this in the revised manuscript. This is now included in Figure S4D and the accompanying figure legends. The MS4A1 axons converge onto a single glomerulus on the

medial ventral posterior side of the olfactory bulb. Therefore, we think the Ms4a1 glomerulus resides where class II odorant receptors form their glomeruli.

5. We would be curious to read more about the authors' ideas on why many specialized subsystems (VNO, TAARs, MS4A, etc) have evolved for mediating innate behaviors like predator avoidance. Do the authors expect distinct behaviors will segregate into different systems? Are they redundant? Is there evidence that the rodent olfactory system has evolved multiple receptors more often for ethologically important odors to confer sensory and behavioral robustness/degeneracy?

We have included our thoughts on this into the discussion of the revised manuscript. We don't know whether there is an enrichment of receptor abundance for ethologically relevant odors as the deorphanization of most receptors has not yet been accomplished, but we think that it makes sense that this could be the case.

REVIEWERS' COMMENTS

Reviewer #1 (Remarks to the Author):

The authors have addressed all my concerns. I now recommend the study for publication.

Reviewer #2 (Remarks to the Author):

The revised manuscript has addressed all concerns, and I recommend accepting it. I have one minor comment: the authors seem to use "odorant receptor" and "olfactory receptor" inconsistently. I suggest using "odorant receptor" when referring to the family originally discovered by Buck and Axel, and "olfactory receptor" for chemosensory receptors expressed in the OSNs more generally.

Reviewer #3 (Remarks to the Author):

The revised manuscript details an interesting and carefully executed study that demonstrates that MS4As function in vivo in the rodent olfactory system as odorant receptors. This result represents a substantive advance for olfactory neuroscience. The authors have answered all questions and concerns that I think can be reasonably expected to be addressed during the review process, and I support publication of the manuscript.

I would gently suggest that the authors would choose a different word than "oblivious" in line 178, but this is stylistic and completely up to them.

We would like to thank all three reviewers for their thoughtful comments on our manuscript. We are all too aware of the many responsibilities that they have, and we are grateful to them for taking the time to provide us feedback on our study. We are pleased that all three reviewers viewed our study so positively, and we addressed the last minor concerns via textual change. Please find a point-by-point response to each of their comments embedded below:

Reviewer #1 (Remarks to the Author):

The authors have addressed all my concerns. I now recommend the study for publication.

We are pleased that the reviewer is satisfied with our revision.

Reviewer #2 (Remarks to the Author):

The revised manuscript has addressed all concerns, and I recommend accepting it. I have one minor comment: the authors seem to use "odorant receptor" and "olfactory receptor" inconsistently. I suggest using "odorant receptor" when referring to the family originally discovered by Buck and Axel, and "olfactory receptor" for chemosensory receptors expressed in the OSNs more generally.

We are pleased that the reviewer is satisfied with our revision. We have changed this throughout the entire manuscript. We thank the reviewer for the suggestion.

Reviewer #3 (Remarks to the Author):

The revised manuscript details an interesting and carefully executed study that demonstrates that MS4As function in vivo in the rodent olfactory system as odorant receptors. This result represents a substantive advance for olfactory neuroscience. The authors have answered all questions and concerns that I think can be reasonably expected to be addressed during the review process, and I support publication of the manuscript.

I would gently suggest that the authors would choose a different word than "oblivious" in line 178, but this is stylistic and completely up to them.

We are pleased that the reviewer is satisfied with our revision. We have changed this phrasing. We thank the reviewer for the suggestion.